# Diffusing States and Matching Scores: A New Framework for Imitation Learning

**Runzhe Wu**
Cornell University
rw646@cornell.edu

**Yiding Chen**
Cornell University
yc2773@cornell.edu

**Gokul Swamy**
Carnegie Mellon University
gswamy@andrew.cmu.edu

**Kianté Brantley**
Harvard University
kdbrantley@g.harvard.edu

**Wen Sun**
Cornell University
ws455@cornell.edu

## Abstract

Adversarial Imitation Learning is traditionally framed as a two-player zero-sum game between a learner and an adversarially chosen cost function, and can therefore be thought of as the sequential generalization of a Generative Adversarial Network (GAN). However, in recent years, diffusion models have emerged as a non-adversarial alternative to GANs that merely require training a score function via regression, yet produce generations of higher quality. In response, we investigate how to lift insights from diffusion modeling to the sequential setting. We propose diffusing states and performing *score-matching* along diffused states to measure the discrepancy between the expert's and learner's states. Thus, our approach only requires training score functions to predict noises via standard regression, making it significantly easier and more stable to train than adversarial methods. Theoretically, we prove first- and second-order instance-dependent bounds with linear scaling in the horizon, proving that our approach avoids the compounding errors that stymie offline approaches to imitation learning. Empirically, we show our approach outperforms both GAN-style imitation learning baselines and discriminator-free imitation learning baselines across various continuous control problems, including complex tasks like controlling humanoids to walk, sit, crawl, and navigate through obstacles.

## 1 Introduction

Fundamentally, in imitation learning (IL, Osa et al. (2018)), we want to match the sequential behavior of an expert demonstrator. Different notions of what matching should mean for IL have been proposed in the literature, from $f$-divergences (Ho & Ermon, 2016; Ke et al., 2021) to Integral Probability Metrics (IPMs, Müller (1997); Sun et al. (2019); Kidambi et al. (2021); Swamy et al. (2021); Chang et al. (2021); Song et al. (2024)). To compute the chosen notion of divergence from the expert demonstrations so that the learner can then optimize it, it is common to train a *discriminator* (i.e. a classifier) between expert and learner data. This discriminator is then used as a reward function for a policy update, an approach known as *inverse reinforcement learning* (IRL, Abbeel & Ng (2004); Ziebart et al. (2008)). Various losses for training discriminators have been proposed (Ho & Ermon, 2016; Kostrikov et al., 2019; Fu et al., 2017; Ke et al., 2021; Swamy et al., 2021; Chang et al., 2024), and IRL has been applied in real life in domains like routing (Barnes et al., 2023) and LLM fine-tuning (Wulfmeier et al., 2024).

As observed by Finn et al. (2016), inverse RL can be seen as the sequential generalization of a Generative Adversarial Network (GAN) (Goodfellow et al., 2020). However, even in the single-shot setting, GANs are known to suffer from issues like unstable training dynamics and mode collapse (Miyato et al., 2018; Arjovsky & Bottou, 2017). In contrast, Score-Based Diffusion Models (Ho et al., 2020; Song & Ermon, 2019; Song et al., 2021) are known to be more stable to train and produce higher quality samples in domains like audio and video (Rombach et al., 2022; Ramesh et al., 2022; Kong et al., 2021). Intuitively, diffusion models corrupt samples from a target distribution with

noise and train a generative model to reverse this corruption. Critically, these models are trained via *score-matching*: a non-adversarial, purely regression-based procedure in which a network is trained to match the score (i.e., gradient of the log probability) of the target distribution.

In fact, prior work has shown that diffusion models are powerful policy classes for IL due to their ability to model multi-modal behavior (Chi et al., 2023). However, these approaches are purely *offline* behavioral cloning (Pomerleau, 1988), and therefore can suffer from the well-known compounding error issue (Ross et al., 2011) that affects *all* offline approaches to IL (Swamy et al., 2021).

Taken together, the preceding points beg the question:

>*Can we lift the insights of diffusion models to inverse reinforcement learning?*

Our answer to this question is SMILING: a new IRL framework that abandons unstable discriminator training in favor of a simpler, score-matching based objective. We first fit a score function to the expert's state distribution. Then, at each iteration of our algorithm, we fit a score function to the state distribution of the mixture of the preceding policies via standard regression-based score matching, before using the combination of these score functions to define a cost function for the policy search step. Our framework treats diffusion model training (score-matching) as a black box which allows us to transfer any advancements in diffusion model training (e.g., better noise scheduling or better training techniques) to IRL. In theory, rather than optimizing either an $f$-divergence or an IPM, this corresponds to minimizing a novel sequential generalization of the Fisher divergence (Johnson, 2004) we term the *Diffusion Score Divergence (DS Divergence)*.

We demonstrate the advantages of our framework in both theory and practice. In theory, we show that our approach can achieve first- and second-order instance-dependent regret bounds, as well as a linear scaling in the horizon to model-misspecification errors arising from expert not being realizable by the learner's policy class or potential optimization error in score-matching and policy search. Thus, we establish that, **by lifting score-matching to IRL, SMILING provably avoids compounding errors and achieves instance-dependent bounds.** Intuitively, the second-order bounds mean that the performance gap between our learned policy and the expert *automatically* shrinks when either the expert or the learned policy has low variance in terms of their performance under the ground-truth reward (e.g. when the expert and dynamics are relatively deterministic). This is because when we perform score-matching, we are actually minimizing the *squared Hellinger distance* between the learner and the expert's state distributions, which has shown to play an important role in achieving second-order regret bounds in the Reinforcement Learning setting (Wang et al., 2024c). The ability to achieve instance-dependent bounds demonstrates the theoretical benefit of Diffusion Score divergence over other metrics such as IPMs and $f$-divergences. In practice, we show that under in the IL from observation only setting (Torabi et al., 2018; Sun et al., 2019), SMILING **outperforms adversarial GAN-based IL baselines, discriminator-free IL baselines, and Behavioral Cloning**[1] **on complex tasks such as controlling humanoids to walk, sit, crawl, and navigate through poles** (see Figure 3). This makes SMILING the first IRL method to solve multiple tasks on the recently released HumanoidBench benchmark (Sferrazza et al., 2024) using only the state information of the expert demonstrations. We release the code for all experiments at `https://github.com/ziqian2000/SMILING`.

## 2 RELATED WORKS

**Inverse Reinforcement Learning.** Starting with the seminal work of Abbeel & Ng (2004), various authors have proposed solving the problem of imitation via inverse RL (Ziebart et al., 2008; Ratliff et al., 2006; Sun et al., 2019; Kidambi et al., 2021; Chang et al., 2021). As argued by Swamy et al. (2021), these approaches can be thought of as minimizing an integral probability metric (IPM) between learner and expert behavior via an adversarial training procedure – we refer interested readers to their paper for a full list. Many other IRL approaches can instead be framed as minimizing an $f$-divergence via adversarial training (Ke et al., 2021), including GAIL (Ho & Ermon, 2016), DAC (Kostrikov et al., 2019), AIRL (Fu et al., 2017), FAIRL (Ghasemipour et al., 2020), f-GAIL (Zhang et al., 2020), and f-IRL (Ni et al., 2021). However, all of the preceding approaches involve training a discriminator, while our technique only requires fitting score functions and does

---

[1]This is despite the fact that BC requires expert actions, while we do not.

not seek to approximate either an $f$-divergence or an IPM. Lai et al. (2024); Wang et al. (2024a) also explored the use of diffusion models for imitation learning. They insert the score matching loss into the $f$-divergence objective of the discriminator. Thus, their method also belongs to the $f$-divergence framework. Huang et al. (2024) also utilize diffusion models for imitation learning, proposing to train a conditional diffusion model as a discriminator for single-step state transitions between the expert and the current policy. Their approach remains within the classic $f$-divergence framework as well. Additionally, various authors have proposed using techniques to either stabilize this training procedure (e.g. via boosting, Chang et al. (2024)) or reducing the amount of interaction required during the RL step (Swamy et al., 2023; Ren et al., 2024; Sapora et al., 2024) – as we focus on improving the reward function used in inverse RL, our approach is orthogonal to and could be naturally applied on top of these techniques.

**Discriminator-Free Inverse Reinforcement Learning.** SQIL (Reddy et al., 2020) replaces training a discriminator with a fixed reward function (+1 for expert data, 0 for learner data). Unfortunately, this means that a performant learner is dis-incentivized to perform expert-like behavior, leading to dramatic drops in performance (Barde et al., 2020). The AdRIL algorithm of Swamy et al. (2021) uses techniques from functional gradients to address this issue, but requires the use off an off-policy RL algorithm to implement, while our framework makes no such assumptions. ASAF (Barde et al., 2020) proposes using the prior policy to compute the optimal $f$-divergence discriminator in closed form, while we focus on a different class of divergences. IQ-Learn (Garg et al., 2021) proposes to perform IRL in the space $Q$ functions, but can therefore suffer from poor performance on problems with stochastic dynamics (Ren et al., 2024) or when the $Q$-function is more complex than the reward function (e.g. navigating to a goal in a maze). Al-Hafez et al. (2023) propose to use a reward regularization to overcome the instabilities of prior work that learns $Q$-functions. Sikchi et al. (2024a) point out that prior methods rely on restrictive coverage assumptions and propose a new method that learns to imitate from arbitrary off-policy data. Jain et al. (2024) propose measuring differences between learner and expert behavior in terms of successor features, but are only able to optimize over deterministic policies. Sikchi et al. (2024b) suggest to learn a multi-step utility function that quantifies the impact of actions on the agent's divergence from the expert's visitation distribution. DRIL (Brantley et al., 2019) relies on ensemble methods and uses the disagreement among models in the ensemble as a surrogate cost function, making it challenging to prove similar theoretical guarantees to our technique. Dadashi et al. (2021) frame IL as minimizing the Wasserstein distance in its primal form rather than the dual commonly used in prior discriminator-based works. Liu et al. (2021) also consider score matching but they use energy-based models instead of diffusion models.

**Diffusion Models for Decision Making.** Diffusion models have been widely studied in the context of robot learning, where diffusion models have demonstrated a strong ability to capture complex and multi-modal data. Recent works have explored using diffusion models to perform behavioral cloning (Chi et al., 2023; Pearce et al., 2023; Chen et al., 2024; Block et al., 2023; Team et al., 2024) and for reward guidance (Nuti et al., 2024) as well as using diffusion models to capture trajectory-level dynamics (Janner et al., 2022; Ajay et al., 2022). We instead focus on lifting insights from diffusion modeling and score matching to IRL, which uses both offline expert data and also online interaction with the environment. Unlike prior diffusion BC work, we use diffusion models for both the expert's and the learner's state distributions, not just the expert's action-conditional distributions. This means we do not require expert action labels, which can be challenging to acquire in practice.

## 3 PRELIMINARIES

### 3.1 MARKOV DECISION PROCESSES

We consider finite-horizon Markov decision processes (MDPs) with state space $\mathcal{S} \subseteq \mathbb{R}^d$, action space $\mathcal{A}$, transition dynamics $P$, (unknown) cost function $c^\star : \mathcal{S} \to \mathbb{R}$,[2] and horizon $H$. The goal is to find a policy $\pi : \mathcal{S} \to \Delta(\mathcal{A})$ that minimizes the expected cost $H \cdot \mathbb{E}_{s \sim d^\pi} c^\star(s)$ where $d^\pi$ denotes the average state distribution induced by the policy $\pi$, i.e., $d^\pi(s) = \sum_{h=1}^{H} \Pr(s_h = s)/H$ where $s_h$ denotes the state at time $h$. The state value function of a policy $\pi$ is defined as $V^\pi(s) = \mathbb{E}_{\tau \sim \pi} \sum_{h=1}^{H} c^\star(s_h)$ where $\tau = (s_1, \dots, s_H)$ denotes the trajectory induced by the policy $\pi$.

---

[2]We assume the cost depends only on states. The extension to state-action costs is straightforward.

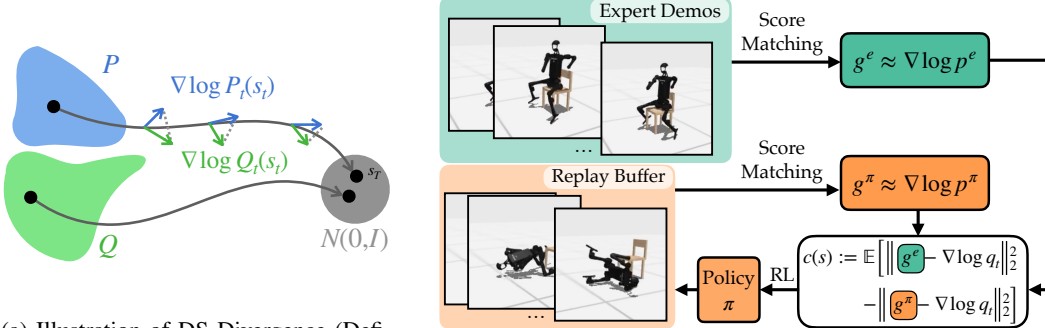

(a) Illustration of DS Divergence (Definition 1).

(b) Illustration of SMILING (Algorithm 1).

Figure 1: Figure (a): The two curves represent the forward diffusion process of distributions $P$ and $Q$. DS Divergence measures the squared difference between the diffusion score functions, $\nabla \log P_t(s_t)$ and $\nabla \log Q_t(s_t)$, along the forward diffusion process of $P$. Figure (b): SMILING first pre-trains a diffusion model from the expert's data. It then iteratively trains diffusion models on learner's data and performs RL to optimize a cost function formed by the learner's score function and the pre-trained expert score function. The cost function is designed to faithfully approximate the DS divergence (Definition 1) between the learner and the expert.

## 3.2 IMITATION LEARNING

In imitation learning, we are usually given expert demonstration in state-action-next-state tuples and aim to learn a policy $\pi$ that mimics the expert policy $\pi^e$ without access to the ground-truth cost function $c^\star$. This paper considers a harder setting where only states are given. Specifically, given a dataset of state demonstrations $\mathcal{D}^e = \{s^{(i)}\}_{i=1}^N$ sampled from expert, we aim to learn a policy $\pi$ that minimizes the discrepancy between its state distributions and the expert's. In addition, we can interact with the environment to collect more (reward-free) learner trajectories.

## 3.3 DIFFUSION MODELS

The *forward process* of a diffusion model adds noise to a sample from the data distribution $p_0$. We can formalize it via a stochastic differential equation (SDE). For simplicity in this paper, we consider the Ornstein-Uhlenbeck (OU) process: $\mathrm{d}\mathbf{x}_t = -\mathbf{x}_t \, \mathrm{d}t + \sqrt{2} \, \mathrm{d}B_t, \mathbf{x}_0 \sim p_0$, where $B_t$ is the standard Brownian motion in $\mathbb{R}^d$. It is known that the *reverse* process $(\mathbf{y}_t)_{t \in [0,T]}$ satisfies the following reverse-time SDE (Anderson, 1982): $\mathrm{d}\mathbf{y}_t = (\mathbf{y}_t + 2\nabla \log p_{T-t}(\mathbf{y}_t)) \, \mathrm{d}t + \sqrt{2} \, \mathrm{d}B_t, \mathbf{y}_0 \sim p_T$ where $U(T)$ denotes the uniform distribution on $[0, T]$, and $B_t$ now denotes reversed Brownian motion. The gradient of the log probability $\nabla \log p_t$ is called the *score function* of the distribution $p_t$. When the score function is known, we can use it to generate samples from $p_0$ by simulating the reverse process. Estimating the score function $\nabla \log p_t$ is called *score matching*, which is typically done via minimizing the regression-based loss: $\min_g \mathbb{E}_{\mathbf{x} \sim p_0} \mathbb{E}_{t \sim U(T)} \mathbb{E}_{\mathbf{x}_t \sim q_t(\cdot \mid \mathbf{x})} \|g(\mathbf{x}_t, t) - \nabla_{\mathbf{x}_t} \log q_t(\mathbf{x}_t \mid \mathbf{x})\|_2^2$. We use $q_t(\mathbf{x}_t \mid \mathbf{x})$ to denote the conditional distribution at time $t$ of the forward process conditioned on the initial state $\mathbf{x}$, which has closed form $q_t(\mathbf{x}_t \mid \mathbf{x}) = \mathcal{N}(\mathbf{x}e^{-t}, (1 - e^{-2t})I)$.

**Applying Diffusion to State Distributions.** For a policy $\pi$, we use $p_t^\pi$ to denote the marginal distribution at time $t$ obtained by applying the forward diffusion process to $d^\pi$, i.e., initial samples are drawn from $p_0^\pi := d^\pi$. When presenting asymptotic results, we use $O(\cdot)$ to hide constants and $\widetilde{O}(\cdot)$ to hide constants and logarithmic factors.

## 4 ALGORITHM

We introduce a novel discrepancy measure between two distributions that our algorithm leverages:

**Definition 1** (Diffusion Score Divergence). *For two distributions $P$ and $Q$, we define the* Diffusion Score Divergence (DS Divergence) *as*

$$D_{\mathrm{DS}}(P, Q) := \mathop{\mathbb{E}}_{s \sim P} \mathop{\mathbb{E}}_{t \sim U(T)} \mathop{\mathbb{E}}_{s_t \sim q_t(\cdot \mid s)} \left\| \nabla \log P_t(s_t) - \nabla \log Q_t(s_t) \right\|_2^2.$$

---

**Algorithm 1** SMILING (**S**core-**M**atching **I**mitation **L**earn**ING**)

---

**Require:** state-only expert demonstration $\mathcal{D}^e = \left\{ s^{(i)} \right\}_{i=1}^N$
1: Estimate score function of expert state distribution:

$$g^e \leftarrow \operatorname*{argmin}_{g \in \mathcal{G}} \, \mathbb{E}_{s \sim \mathcal{D}^e} \, \mathbb{E}_{t \sim U(T)} \, \mathbb{E}_{s_t \sim q_t(\cdot \,|\, s)} \left[ \|g(s_t, t) - \nabla_{s_t} \log q_t(s_t \,|\, s)\|_2^2 \right]$$

2: **for** $k = 1, 2, \ldots, K$ **do**
3:     Estimate the score function of learner state distributions:

$$g^{(k)} \leftarrow \operatorname*{argmin}_{g \in \mathcal{G}} \sum_{i=1}^{k-1} \, \mathbb{E}_{s \sim d^{\pi^{(i)}}} \, \mathbb{E}_{t \sim U(T)} \, \mathbb{E}_{s_t \sim q_t(\cdot \,|\, s)} \left[ \|g(s_t, t) - \nabla_{s_t} \log q_t(s_t \,|\, s)\|_2^2 \right].$$

4:     Update policy $\pi^{(k)}$ via RL (e.g., SAC) on cost $c^{(k)}$ (Eq. 3): $\pi^{(k)} \leftarrow \mathrm{RL}(c^{(k)})$
5: **end for**

---

*Here $q_t(\cdot \,|\, s)$ represents the conditional probability of the forward diffusion process at time $t$ conditioned on the initial state $s$; $P_t$ and $Q_t$ denote the marginal distributions at time $t$ obtained by applying the forward diffusion process to $P$ and $Q$, respectively. We call $\nabla \log P_t$ and $\nabla \log Q_t$ the* Diffusion Score Function *of $P$ and $Q$.*

Figure 1a illustrates the DS divergence. It measures the difference between two distributions by comparing their diffusion score functions within one diffusion process. It is analogous to Fisher divergence but differs by incorporating an expectation over the diffusion process. DS divergence is a strong divergence in the sense that, whenever the DS divergence between the two distributions is small, the KL divergence, Hellinger distance, and total variation distance are all small (see Lemma 2 and also Chen et al. (2023b); Oko et al. (2023)).

In our algorithmic framework **S**core-**M**atching **I**mitation **L**earn**ING** — SMILING, *we propose to frame imitation learning as the minimization of the DS divergence between the expert's and the (history of) learner state distributions.* We begin by discussing how to do so before discussing the theoretical benefits of doing so.

The first step is to *pre-train* a score function estimator $g^e(s, t)$ on expert's state distribution $\nabla_s \log p_t^{\pi^e}(s)$. This can be done via standard least-squares regression-based score-matching, as shown in Line 1. Here $\mathcal{G}$ denotes the function class for score estimators. Then, we seek to find a policy $\pi$ that minimizes the DS divergence between learner and expert state distributions:

$$\ell(\pi) := \mathbb{E}_{s \sim d^\pi, t \sim U(T), s_t \sim q_t(\cdot | s)} \|g^e(s_t, t) - \nabla_{s_t} \log p_t^\pi(s_t)\|_2^2, \tag{1}$$

where we have approximated the expert's diffusion score function by $g^e(s_t, t)$. However, $\ell(\pi)$ is not directly computable since we do not know the learner's score function $\nabla_{s_t} \log p_t^\pi(s_t)$. A naive approach would be to directly learn an estimator for $\nabla_{s_t} \log p_t^\pi(s_t)$ via score matching and substitute it into the equation. However, as we prove in Appendix B, this can introduce significant errors due to the unboundedness of a difference of score functions and variance-related concerns. We now derive a method that does not suffer from this issue.

Recall that we defined $q_t(s_t \,|\, s)$ as the distribution of $s_t$ given the initial sample $s$, which is a simple Gaussian distribution with an appropriately scaled variance. To develop our method, we first note that, luckily, $\nabla_{s_t} \log q_t(s_t|s)$ is an unbiased estimator of the score $\nabla_{s_t} \log p_t^\pi(s_t)$, because $\mathbb{E}_{s|s_t}[\nabla_{s_t} \log q_t(s_t|s)] = \nabla_{s_t} \log p_t^\pi(s_t)$ (Song et al., 2021). Given this, perhaps the most immediate strategy would be to simply replace $\nabla_{s_t} \log p_t^\pi(s_t)$ by its unbiased estimator, $\nabla_{s_t} \log q_t(s_t|s)$, in Eq. 1. However, in contrast to linear objectives like an IPM, to approximate a squared objective like the DS Divergence accurately, we need to get both the expectation (i.e. have an unbiased estimator) as well as the *variance* correct. If we don't, then $\mathbb{E}_{s \sim d^\pi, t \sim U(T), s_t \sim q_t(\cdot | s)} \|g^e(s_t, t) - \nabla_{s_t} \log q_t(s_t|s)\|_2^2$ will differ from $\ell(\pi)$ by a term related to *variance* of the estimator $\nabla_{s_t} \log q_t(s_t|s)$, i.e.

$$\mathbb{E}_{s \sim d^\pi, t \sim U(T), s_t \sim q_t(\cdot | s)} \|\nabla_{s_t} \log p_t^\pi(s_t) - \nabla_{s_t} \log q_t(s_t|s)\|_2^2. \tag{2}$$

To faithfully approximate $\ell(\pi)$, we need to estimate the variance term and subtract it from $\mathbb{E}_{s \sim d^\pi, t \sim U(T), s_t \sim q_t(\cdot|s)} \|g^e(s_t, t) - \nabla_{s_t} \log q_t(s_t|s)\|_2^2$. We can do this by adding a term to our objective that is minimized at the variance and using another function / network to optimize it. This leads us to the following estimator for $\ell(\pi)$:

$$\hat{\ell}(\pi) := \mathbb{E}_{s \sim d^\pi, t \sim U(T), s_t \sim q_t(\cdot|s)} \|g^e(s_t, t) - \nabla_{s_t} \log q_t(s_t|s)\|_2^2$$
$$- \underbrace{\min_{g \in \mathcal{G}} \mathbb{E}_{s \sim d^\pi, t \sim U(T), s_t \sim q_t(\cdot|s)} \|g(s_t, t) - \nabla_{s_t} \log q_t(s_t|s)\|_2^2}_{\text{Minimized at the variance of Eq. 2}}$$

where $\mathcal{G}$ is the function class for score estimators. Crucially, the highlighted term (orange) in the above expression estimates the variance in Eq. 2 since one of its minimizers will be the Bayes optimal $\nabla_{s_t} \log p_t^\pi(s_t)$, i.e. the conditional expectation $\mathbb{E}_{s|s_t}[\nabla_{s_t} \log q_t(s_t|s)]$.[3] Observe that minimizing this orange term just requires a simple regression-based score matching objective. So, while naive score matching causes issues due to the variance of the estimator, a more clever application of score matching can be used to fix this concern.

With $\hat{\ell}(\pi)$ now serving as a valid approximation of $\ell(\pi)$, the IL problem reduces to searching for a policy to minimize $\hat{\ell}(\pi)$, i.e., $\min_\pi \hat{\ell}(\pi)$. To facilitate this, we define payoff $\mathcal{L}(\pi, g)$ as:

$$\mathcal{L}(\pi, g) := \mathbb{E}_{s \sim d^\pi, t \sim U(T), s_t \sim q_t(\cdot|s)} \left( \|g^e(s_t, t) - \nabla_{s_t} \log q_t(s_t|s)\|_2^2 - \|g(s_t, t) - \nabla_{s_t} \log q_t(s_t|s)\|_2^2 \right).$$

Then, minimizing $\hat{\ell}(\pi)$ is equivalent to solving the two-player zero sum game $\min_\pi \max_g \mathcal{L}(\pi, g)$. To solve this game, we propose following a no-regret strategy over $g$ against a best response over $\pi$ (i.e. a *dual* algorithm (Swamy et al., 2021)). Specifically, Algorithm 1 applies Follow-the-Leader (Shalev-Shwartz et al., 2012) to optimize $g$ in Line 3,[4] and performs best response computation over $\pi$ via RL (Line 4) under cost function $c^k(s)$:

$$c^{(k)}(s) := \mathbb{E}_{t \sim U(T)} \mathbb{E}_{s_t \sim q_t(\cdot|s)} \left[ \left\| g^e(s_t, t) - \nabla_{s_t} \log q_t(s_t|s) \right\|_2^2 - \left\| g^{(k)}(s_t, t) - \nabla_{s_t} \log q_t(s_t|s) \right\|_2^2 \right]. \tag{3}$$

**Remark 1** (Noise-prediction Form of the Cost Function). *The cost function in Eq. 3 involves the score function $\nabla \log q_t(s_t|s)$, which has a closed-form expression for most modern diffusion models including DDPM. Specifically, when the diffusion follows the OU process (described in Section 3.3), the score function takes the form $\nabla \log q_t(s_t|s) = (1 - e^{-2t})^{-1}(se^{-t} - s_t)$, which is exactly the noise added to the original sample $s$ over diffusion (recalling that for the OU process, $q_t(s_t|s) = \mathcal{N}(se^{-t}, (1 - e^{-2t})I))$. We denote this noise by $\epsilon$. Then, the cost function (Eq. 3) is equivalent to $c^{(k)}(s) := \mathbb{E}_{t \sim U(T)} \mathbb{E}_{\epsilon \sim \mathcal{N}(0,I)}[\|g^e(se^{-t} + \epsilon, t) - \epsilon\|_2^2 - \|g^{(k)}(se^{-t} + \epsilon, t) - \epsilon\|_2^2]$. This closely resembles the noise prediction in DDPM.*

To implement FTL, in Line 3, we use the classic idea of Data Aggregation (DAgger) (Ross et al., 2011), which corresponds to aggregating all states collected from prior learned policies $\pi^{(1)}, \ldots, \pi^{(k-1)}$, and perform a score-matching least square regression on the aggregated dataset. The RL procedure can take advantage of any modern RL optimizers. In our experiments, we use SAC (Haarnoja et al., 2018) and DreamerV3 (Hafner et al., 2023), which serve as representative model-free and model-based RL algorithms.

## 5 THEORETICAL RESULTS

Our goal in this section is to demonstrate that SMILING **achieves instance-dependent regret bounds while at the same time avoid compounding errors** when model misspecification, optimization error, and statistical error exist. In practice, we can only estimate the diffusion score

---

[3]Note that similar ideas have been used in the offline RL literature for designing min-max based algorithms for estimating value functions (e.g., Chen & Jiang (2019); Uehara et al. (2021)).

[4]Note that $\mathcal{L}(\pi, g)$ is a square-loss functional with respect to $g$. Since square loss is strongly convex, FTL is no-regret. In contrast, when optimizing an IPM, one has to use Follow-the-Regularized-Leader (FTRL) to have the no-regret property (e.g., Sun et al. (2019); Swamy et al. (2021)), making implementation more complicated.

function up to some statistical error or optimization error due to finite samples. For similar reasons, the RL procedure (Line 4) can only find a near-optimal policy especially when the policy class is not rich enough to capture the expert's policy. To demonstrate that our algorithm can tolerate these errors, we explicitly study these errors in our theoretical analysis, partically how our regret bound scales with respect to these errors. We provide the following assumptions to formalize these errors.

**Assumption 1.** *We have the following error bounds, corresponding respectively to Lines 1, 3 and 4 of Algorithm 1:*

*(a) The estimator $g^e$ is accurate up to some error $\epsilon_{\text{score}}$: $\mathbb{E}_{s \sim d^{\pi^e}} \mathbb{E}_{t \sim U(T)} \mathbb{E}_{s_t \sim q_t(\cdot \mid s)}[\|g^e(s_t, t) - \nabla_{s_t} \log p_t^{\pi^e}(s_t)\|_2^2] \leq \epsilon_{\text{score}}^2$;*

*(b) There exists a function $\text{Regret}(K)$ sublinear in $K$ such that the sequence of score function estimators $\{g^{(k)}\}_{k=1}^K$ has regret bounded by $\text{Regret}(K)$:*

$$\sum_{k=1}^K \mathbb{E}_{s \sim d^{\pi^{(k)}}} \mathbb{E}_{t \sim U(T)} \mathbb{E}_{s_t \sim q_t(\cdot \mid s)} \left[ \left\| g^{(k)}(s_t, t) - \nabla_{s_t} \log q_t(s_t \mid s) \right\|_2^2 \right]$$

$$\leq \min_g \sum_{k=1}^K \mathbb{E}_{s \sim d^{\pi^{(k)}}} \mathbb{E}_{t \sim U(T)} \mathbb{E}_{s_t \sim q_t(\cdot \mid s)} \left[ \|g(s_t, t) - \nabla_{s_t} \log q_t(s_t \mid s)\|_2^2 \right] + \text{Regret}(K);$$

*(c) There exists $\epsilon_{\text{RL}} > 0$ such that, for all $k = 1, \ldots, K$, the RL procedure finds an $\epsilon_{\text{RL}}$-optimal policy within some function class $\Pi$: $\mathbb{E}_{s \sim d^{\pi^{(k)}}} c^{(k)}(s) - \min_{\pi \in \Pi} \mathbb{E}_{s \sim d^{\pi}} c^{(k)}(s) \leq \epsilon_{\text{RL}}$. Note that we do not assume $\pi^e \in \Pi$.*

**Item (a)** is a standard assumption in diffusion process and score matching (Chen et al., 2023b;a; Lee et al., 2022). When the function class $\mathcal{G}$ (hypothesis space) is finite, $\epsilon_{\text{score}}^2$ typically scales at a rate of $\widetilde{O}(\ln(|\mathcal{G}|)/N)$ in terms of the number of samples $N$. For an infinite function class, more advanced bounds can be established (e.g., see Theorem 4.3 in Oko et al. (2023)). We emphasize that the discretization error arising from approximating the diffusion SDE using Markov chains is included in $\epsilon_{\text{score}}$ as well. **Item (b)** is similar to Item (a) but for the sequence of score function estimators. It can be satisfied by applying Follow-the-Leader (FTL) since the loss functional for $g$ is a square loss. Other no-regret algorithms, such as Follow-the-regularized-leader (FTRL) or online gradient descent, can also be used. Under similar conditions as in Item (a), one typically achieves $\text{Regret}(K) = \widetilde{O}(\sqrt{K})$. **Item (c)** can be satisfied as long as an efficient RL algorithm is applied at each iteration in Line 4. The rate of $\epsilon_{\text{RL}}$ has been well-studied in the RL theory literature for various of MDPs. For instance, in tabular MDPs, one can achieve $\epsilon_{\text{RL}} = \widetilde{O}(\sqrt{SA/M})$ where $S$ is the number of states, $A$ is the number of actions, and $M$ is the number of RL rollout samples. We emphasize that $\epsilon_{\text{RL}}$ in this case does not scale with $H$ as it is defined as the upper bound on the *averaged* return instead of the cumulative return over $H$ steps.

In the theoretical analysis, we will not assume any misspecification of score functions. Specifically, for any policy $\pi$, we assume that its score function lies in the function class $\mathcal{G}$ (i.e., $\nabla \log p_t^\pi \in \mathcal{G}$). However, we allow for misspecification in the RL procedure (i.e., $\pi^e$ may not be in $\Pi$). This enables us to focus more on the misspecification of the RL procedure under our defined framework.

Next, we introduce the regularity assumptions on the diffusion process, which are standard in the literature on diffusion models.

**Assumption 2.** *For all $\pi$, the following two conditions hold: (1) the diffusion score function $\nabla \log p_t^\pi$ is Lipschitz continuous with a finite constant for all $t \in [0, T]$, and (2) the second moment of $d^\pi$ is bounded: $\mathbb{E}_{s \sim d^\pi} \left[ \|s\|_2^2 \right] \leq m$ for some $m > 0$.*

The first condition ensures that the score function behaves well so we can transfer the DS divergence bound to a KL divergence bound. The second condition is needed for the exponential convergence of the forward diffusion process (i.e., the convergence to the standard Gaussian in terms of KL divergence). We note that the second condition is readily satisfied in certain simple cases such as when the state space is bounded. Notably, if we disregard discretization errors, the Lipschitz constant will not appear in our theoretical results as long as it is finite, even if it is arbitrarily large. Similarly, the constant $m$ only appears within logarithmic factors so it can be exponentially large.

However, when discretization errors are considered, both the Lipschitz constant and $m$ will appear in $\epsilon_{\text{score}}$ and $\text{Regret}(T)$ (see, e.g., Chen et al. (2023b;a); Oko et al. (2023)).

Now we are ready to present our main theoretical result. Let $\pi^{(1:K)}$ denote the mixture policy of $\{\pi^{(1)}, \ldots, \pi^{(K)}\}$. Particularly, $\pi^{(1:K)}$ is executed by first choosing a policy $\pi^{(k)}$ uniformly at random and then executing the chosen policy. For any policy $\pi$, we define the variance of its return as $\text{Var}^\pi := \text{Variance}(\sum_{h=1}^H c^\star(s_h))$. Below is our main theoretical result with proof in Appendix C.

**Theorem 1.** *Under Assumptions 1 and 2, Algorithm 1 achieves these instance-dependent bounds:*

$$\text{(Second-order)} \qquad V^{\pi^{(1:K)}} - V^{\pi^e} = \widetilde{O}\left(\sqrt{\min\left(\text{Var}^{\pi^e}, \text{Var}^{\pi^{(1:K)}}\right) \cdot \epsilon} + \epsilon H\right);$$

$$\text{(First-order)} \qquad V^{\pi^{(1:K)}} - V^{\pi^e} = \widetilde{O}\left(\sqrt{\min\left(V^{\pi^e}, V^{\pi^{(1:K)}}\right) \cdot \epsilon H} + \epsilon H\right)$$

*where we define total error* $\epsilon := \begin{cases} \epsilon_{\text{score}}^2 + \epsilon_{\text{RL}} + \text{Regret}(K)/K & \text{if } \pi^e \in \Pi, \\ \epsilon_{\text{score}}^2 + \epsilon_{\text{RL}} + \text{Regret}(K)/K + \epsilon_{\text{mis}} & \text{if } \pi^e \notin \Pi, \end{cases}$ *and the mis-specification* $\epsilon_{\text{mis}} := \min_{\pi \in \Pi} \ell(\pi)$ *where we recall that* $\ell(\cdot)$ *is defined in Eq. 1.*

Here the misspecification error $\epsilon_{\text{mis}}$ measures the minimum possible DS divergence to the pre-trained expert score $g^e$. We note that $\epsilon_{\text{mis}}$ is algorithmic-path independent once given $g^e$, i.e., it does not depend on what the algorithm does — it is a quantity that is pre-determined when the IL problem is formalized and the expert score $g^e$ is pre-trained. If we increase the expressiveness of $\Pi$, $\epsilon_{\text{mis}}$ will decrease. In contrast, interactive IL algorithms DAgger (Ross et al., 2011) and AggreVate(D) (Ross & Bagnell, 2014; Sun et al., 2017), which claim to have no compounding errors, actually have algorithmic-path dependent misspecification errors. This means their misspecification error does not necessarily decrease and in fact can increase when one increases the capacity of $\Pi$ (since it affects the algorithm's behavior).

It is known that the second-order bound is tighter and subsumes the first-order bound (Wang et al., 2024c). Importantly, previous results on second-order bounds for sequential decision-making have relied exclusively on the Maximum Likelihood Estimator (MLE) (Wang et al., 2023; 2024c; Foster et al., 2024; Wang et al., 2024b; 2025), which in general is not computationally tractable even for simple exponential family distributions (Pabbaraju et al., 2024). In this work, we show for the first time that a computationally tractable alternative — diffusion score matching — can achieve these instance-dependent bounds in the context of IRL. In particular, our second-order bounds (or the first-order bound) scale with the *minimum* of $\text{Var}^{\pi^e}$ and $\text{Var}^{\pi^{(1:K)}}$ (or minimum of $V^{\pi^e}$ and $V^{\pi^{(1:K)}}$). Hence, our bounds are sharper whenever one of the two is small. In contrast, bounds in these prior RL work do not scale with the min of variances associated with the learned policies and the comparator policy $\pi^e$. This subtlety is likely due to the difference between the IRL setting and the general RL setting instead of techniques in the analysis.

**Improved sample efficiency with respect to the size of expert data.** Let's now focus on sample complexity with respect to expert data, which is typically the most expensive data to collect in IL, to highlight the benefits of our method over previous approaches. For conciseness, we ignore all errors that are purely related to computation. Specifically, we set $\epsilon_{\text{RL}} \to 0$, as it only depends on the time spent to run the RL procedure, let the number of iterations $K \to \infty$ so $\text{Regret}(K)/K \to 0$, and assume $\pi^e \in \Pi$. Under these conditions, our bound solely depends on $\epsilon_{\text{score}}$. For simplicity, we further assume a finite score function class $\mathcal{G}$, neglect misspecification errors (i.e., $\nabla \log p_t^\pi(x) \in \mathcal{G}$ for all $\pi$), and assume the magnitude of the score loss is $O(1)$. Now standard regression analysis (Agarwal et al., 2019; Oko et al., 2023) yields $\epsilon_{\text{score}} = O(\sqrt{\log(|\mathcal{G}|)/N})$ where $N$ is the number of expert samples. Plugging this back into our second-order bound, we obtain

$$\frac{1}{K} \sum_{k=1}^K V^{\pi^{(k)}} - V^{\pi^e} = O\left(\sqrt{\min\left(\text{Var}^{\pi^e}, \text{Var}^{\pi^{(1:K)}}\right) \cdot \frac{\log(|\mathcal{G}|)}{N}} + \frac{H \cdot \log(|\mathcal{G}|)}{N}\right).$$

When the expert's cost or the learned policy's cost has low variance, i.e., $\text{Var}^{\pi^e} \to 0$ or $\text{Var}^{\pi^{(1:K)}} \to 0$, our bound simplifies to $O(H \cdot \log(|\mathcal{G}|)/N)$. This shows a significant improvement over prior sample complexity bounds for IRL with general function approximation, which are

typically $O(H\sqrt{\log(|\mathcal{G}|)/N})$ (e.g., IPM-based methods (Kidambi et al., 2021; Chang et al., 2021)). This demonstrates the benefit of using diffusion processes and score matching over IPMs.

In addition to the sample benefit above, we show our approach can require less expressive function approximators than discriminator-based methods. Please refer to Appendix A for more details.

## 6 EXPERIMENTS

We evaluate our method, `SMILING`, on a set of continuous control tasks from the Deepmind Control Suite (Tassa et al., 2018) and HumanoidBench (Sferrazza et al., 2024). These tasks vary in difficulty, as summarized in Table 1 of Appendix D. While we will compare to multiple baselines including non-adversarial baselines, we focus on testing our hypothesis that score-matching based approach should outperform $f$-divergence-based approach. Additional results and details are in Appendix D.

**Baselines.** We include Discriminator Actor-Critic (DAC), IQ-Learn, DiffAIL, and Behavior Cloning (BC) as baselines. Our DAC implementation is based on the original design by Kostrikov et al. (2019) that uses JS-divergence GAN-style objective. We use the official implementation of IQ-Learn. We tried several configurations provided in their implementation for the MuJoCo tasks and reported the best one. DiffAIL also follows official implementation. The BC performance is reported as the maximum episode reward obtained across all checkpoints during training. We found MLE outperforms squared loss for BC, so all BC results below use MLE.

**Diffusion Models.** We employ the Denoising Diffusion Probabilistic Model (DDPM) as the diffusion model. The score function is modeled via an MLP with one hidden layer of 256 units, and we use the same architecture for the discriminator in DAC. The diffusion process is discretized into 5K steps and we use a learnable time embedding of 16 dimensions. We approximate the cost function (Eq. 3) via empirical mean using 500 samples. Since the scale of the score matching loss (and consequently our cost function) is sensitive to the data scale (in this context, the scale of state vector), we normalize the cost of each batch to have zero mean and a standard deviation of 0.1.

**RL Solvers.** For DMC tasks, we use Soft Actor-Critic (SAC) (Haarnoja et al., 2018) as the RL solver for both DAC and our method, based on the implementation provided by Yarats & Kostrikov (2020). For HumanoidBench tasks, we adopt DreamerV3 (Hafner et al., 2023) using the implementation from the HumanoidBench repository (Sferrazza et al., 2024). Additionally, we maintain a separate "state buffer" that operates the same as the replay buffer but only stores states. This buffer is used to sample states to update the score function in our method and the discriminator in DAC.

**Overall, we keep the implementation of `SMILING` and the baseline DAC as close as possible (e.g., using the same RL solver) so they only differ in objective functions** (i.e., `SMILING` learns score functions while DAC learns discriminators to approximate JS divergence). This allows us to show the benefits of our score-matching approach to the discriminator-based GAN-style ones.

**Expert Policy and Demonstrations.** We use the aforementioned RL solvers to train expert policies and collect demonstrations by running the expert policy for five episodes. Each task has a fixed 1K-step horizon without early termination, and thus the resulting dataset has 5K steps per task.

### 6.1 LEARNING FROM STATE-ONLY DEMONSTRATIONS

Figure 2 illustrates the results across six control tasks when the expert demonstrations are state-only. We trained for 0.4M steps on `Ball-in-cup-catch` and for 3M steps on all other tasks. Our method matches expert performance and outperforms all baselines on five of them. Specifically, our method outperforms all baselines by a large margin on four. For `humanoid-sit`, while none achieves expert performance, ours converges the fastest. For `cheetah-run`, although DAC initially shows faster convergence, it becomes unstable soon and oscillates significantly after 2M steps; in contrast, our algorithm remains much more stable and surpasses DAC in 3M steps. Notably, we observed DiffAIL collapses at the beginning of training on all four humanoid tasks due to numerical issues (neural network weights diverging to infinity). Hence, we were unable to report the those results. Also note that, in the plot, we include BC as a reference though it is not directly comparable since it uses state-action demonstrations. Here BC serves as a performance upper bound for Behavioral Cloning from Observation (BCO) (Torabi et al., 2018), which infers the expert's actions from state-only data before training. Nevertheless, we observe that both our method and DAC consistently outperform

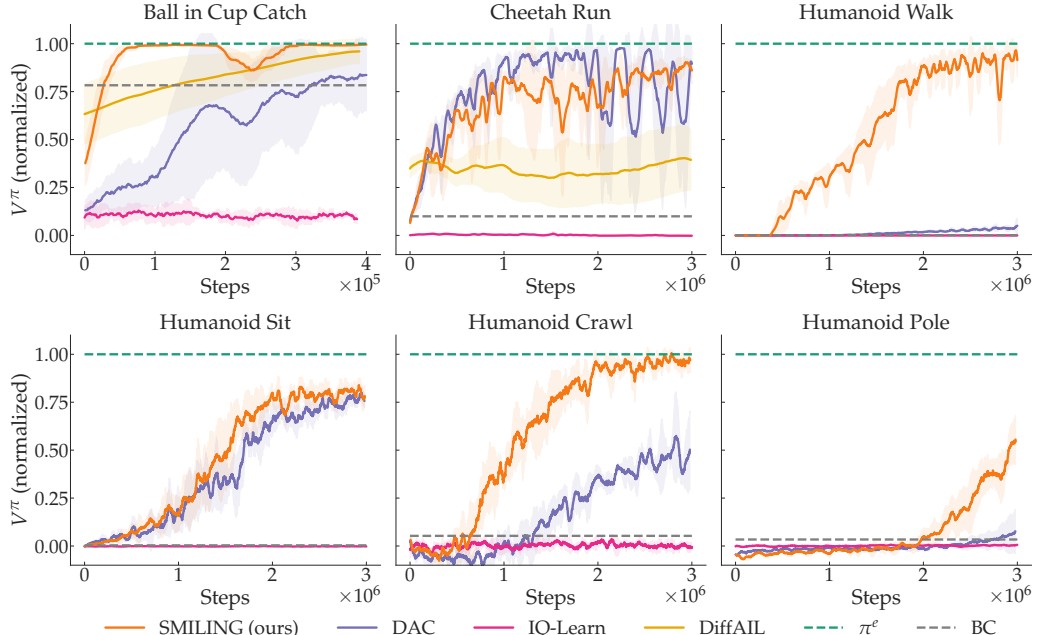

Figure 2: Learning curves for learning from state-only data across five random seeds. The x-axis corresponds to the number of environment steps (also the number of policy updates). The y-axis is normalized such that the expert performance is one and the random policy is zero. Our method clearly outperforms all baselines in five tasks out of six.

BC (and thus, BCO). We provide video demos of `SMILING` and DAC on the `humanoid-crawl` and `humanoid-pole` tasks in Figure 3 to give a clear visual comparison of their performance differences.

## 6.2 ADDITIONAL RESULTS

**Learning from State-Action Demonstrations.** Our method can be easily extended to learning from state-action demonstrations by appending the action vector to the state vector for both training and computing rewards. Hence, we also explore its performance in this setting. The results are consistent with the state-only setting and can be found in Appendix D.1.

**Number of Expert Demonstrations.** We also investigate the impact of the number of expert demonstrations on the performance of our method and baselines in order to support our theoretical results in Section 5. The results can be found in Appendix D.2 and show that our method is clearly more sample-efficient and performs well when the number of expert demonstrations is small.

**Expressive of Diffusion Models.** In Appendix A, we argued that score matching is more expressive than discriminator-based methods, which may explain why our approach generally outperforms DAC. To support it, we conducted an ablation study on the `cheetah-run` task where ours converged slower than DAC previously. We removed the activation functions in both the discriminator of DAC and the score function of our method to make them *purely linear*. We found that DAC's performance degrades notably while our method remains more effective and outperforms DAC clearly. Detailed results are in Appendix D.3. This reinforces our claim about the expressiveness of score matching.

## 7 CONCLUSION

We propose a new IL framework, Score-Matching Imitation Learning (`SMILING`), that leverages the diffusion score function to learn from expert demonstrations. Unlike previous methods, `SMILING` is not formulated from any $f$-divergence or IPM perspective. Instead, it directly matches the score of the learned policy to that of the expert. Theoretically, we show that score function is more expressive than learning a discriminator on $f$-divergence. Additionally, our method achieves first- and second-order bounds in sample complexity, which are instance-dependent and tighter than previous results. In practice, our method outperforms existing algorithms on a set of continuous control tasks.

ACKNOWLEDGMENTS

GKS was supported in part by a STTR grant. WS acknowledges support from NSF IIS-2154711, NSF CAREER 2339395, and DARPA LANCER: LeArning Network CybERagents. This work has been made possible in part by a gift from the Chan Zuckerberg Initiative Foundation to establish the Kempner Institute for the Study of Natural and Artificial Intelligence.

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

## A    COMPARISON TO DISCRIMINATOR-BASED (f-DIVERGENCE AND IPM) METHODS

In this section, we compare DS divergence with $f$-divergence, particularly its discriminator-based approximation (the key metric used by IL algorithms such as GAIL and DAC):

$$\max_{f \in \mathcal{F}} \ \mathbb{E}_{s \sim p} \left[ \log f(s) \right] + \mathbb{E}_{s \sim q} \left[ \log(1 - f(s)) \right]$$

where $\mathcal{F} \subseteq (0, 1)^{\mathcal{S}}$ is a restricted function class. It is known that the above is a lower bound of the Jensen-Shannon (JS) divergence between $p$ and $q$. In particular, when $p/(p + q) \in \mathcal{F}$, it is exactly equal to the JS divergence. However, representing $p/(p + q)$ may be challenging. Below we show why score-matching is more preferred than discriminator-based $f$-divergence via the example of exponential family distributions,

**On the expressiveness of score functions and discriminators.**    Consider the distributions $p$ and $q$ to be from the exponential family, $p(s) = \exp(w^T \phi(s))/Z_p$ (and similar for $q$), where $\phi$ is a quadratic feature map, and $Z_p$ is the partition function. In this case, the score function $\nabla_s \log p(s) = \nabla_s \phi(s)$ is simply linear in $s$. In contrast, the optimal discriminator for JS divergence of $p$ and $q$ has the form of $p/(p+q)$, which is inherently nonlinear and cannot be captured by a linear function on $s$. As a result, if we use linear functions to model discriminators, the discriminator-based objective cannot even faithfully serve as a tight lower bound for the JS divergence. Thus minimizing a loose lower bound of the JS-divergence does not imply minimizing the JS-divergence itself.

Now consider integral probability metric (IPM), i.e., $\max_{f \in \mathcal{F}} (\mathbb{E}_{s \sim q} f(s) - \mathbb{E}_{s \sim p} f(s))$. When $p$ and $q$ are from the exponential family, to perfectly distinguish $q$ and $p$, one needs to design a discriminator in the form of $\theta^\top \phi(x)$ since $\phi$ is the sufficient statistics of the exponential family distribution. In our example, $\phi$ is a quadratic feature mapping, meaning that we need a non-linear discriminator for IPM as well.

Hence, score matching only needs relatively weaker score function class to represent complicated distributions to control certain powerful divergence including KL divergence and Hellinger distance. However, for $f$-divergence-based or IPM-based methods, we need a more expressive discriminator function class to make them serve as a tight lower bound of the divergences. This perhaps explains why in practice for generative models, score matching based approach in general outperforms more traditional GAN based approach. In our experiment, we conduct an ablation study in Appendix D.3 to compare our approach to a GAN-based baseline under linear function approximation for score functions and discriminators, and we show that `SMILING` can still learn well even when the score functions are linear with respect to the state $s$.

**Can $f$-divergence or IPM-based approaches achieve second-order bounds?**    Wang et al. (2024b) has demonstrated that simply matching expectations (first moment) is insufficient to achieve second-order bounds in a simpler supervised learning setting. Mapping this to the IL setting, if the true cost is linear in feature $\phi(s)$, i.e., $c^\star(s) = \theta^\star \cdot \phi(s)$, and we design discriminator $g(s) = \theta^\top \phi(s)$, then IL algorithm that learns a policy $\pi$ to only match policy's expected feature $\mathbb{E}_{s \sim \pi} \phi(s)$ to the expert's $\mathbb{E}_{s \sim \pi^e} \phi(s)$ cannot achieve a second-order regret bound. For f-divergence such as JS-divergence, unless the discriminator class is rich enough to include the optimal discriminator, the discriminator-based objective is only a loose lower bound of the true $f$-divergence. On the other hand, making the discriminator class rich enough to capture the optimal discriminator can increase both statistical and computational complexity for optimizing the discriminator. Taking the exponential family distribution as an example again, there the optimal discriminator relies on the partition functions $Z_q$ and $Z_p$ which can make optimization intractable (Pabbaraju et al., 2024).

## B    ERROR ANALYSIS OF OBJECTIVE

In this section, we show the following two things: (1) why directly replacing the diffusion score function of $\pi$ with an estimator $g^\pi$ in $\ell(\pi)$ (Eq. 1) incurs large error; (2) why the error is small in our derived objective $\widehat{\ell}(\pi)$.

On the one hand, replacing the score function of $\pi$ with $g^\pi$ incurs the following error:

$$\underset{s \sim d^\pi}{\mathbb{E}} \underset{t \sim U(T)}{\mathbb{E}} \underset{s_t \sim q_t(\cdot \mid s)}{\mathbb{E}} \left[ \|g^e(s_t, t) - g^\pi(s_t)\|_2^2 - \|g^e(s_t, t) - \nabla_{s_t} \log p_t^\pi(s_t)\|_2^2 \right]$$

$$= \underset{s \sim d^\pi}{\mathbb{E}} \underset{t \sim U(T)}{\mathbb{E}} \underset{s_t \sim q_t(\cdot \mid s)}{\mathbb{E}} \left[ \underbrace{(\nabla_{s_t} \log p_t^\pi(s_t) - g^\pi(s_t))}_{(a)} \underbrace{(2g^e(s_t, t) - g^\pi(s_t) - \nabla_{s_t} \log p_t^\pi(s_t))}_{(b)} \right].$$

Here the scale of term (a) is bounded by the statistical error from score matching. However, the scale of term (b) is unbounded since $g^e$ can arbitrarily deviate from both $g^\pi$ and $\nabla_{s_t} \log p_t^\pi(s_t)$. Hence, the error is potentially unbounded.

On the other hand, our derived objective is bounded. To see this, we note that our objective incur the following error:

$$\underset{s \sim d^\pi}{\mathbb{E}} \underset{t \sim U(T)}{\mathbb{E}} \underset{s_t \sim q_t(\cdot \mid s)}{\mathbb{E}} \left[ \|\nabla_{s_t} \log p_t^\pi(s_t) - \nabla_{s_t} \log q_t(s_t \mid s)\|_2^2 - \|g^\pi(s_t) - \nabla_{s_t} \log q_t(s_t \mid s)\|_2^2 \right]$$

$$= \underset{s \sim d^\pi}{\mathbb{E}} \underset{t \sim U(T)}{\mathbb{E}} \underset{s_t \sim q_t(\cdot \mid s)}{\mathbb{E}} \left[ \|\nabla_{s_t} \log p_t^\pi(s_t) - g^\pi(s_t)\|_2^2 \right].$$

Here the equality is by Lemma 4. We observe that it is exactly bounded by the statistical error from score matching.

## C  PROOF OF THEOREM 1

### C.1  SUPPORTING LEMMAS

The following lemma is from Wang et al. (2024c) and the fact that triangular discrimination is equivalent to the squared Hellinger distance up to a multiplicative constant: $2D_H^2 \leq D_\triangle \leq 4D_H^2$ (Topsoe, 2000).

**Lemma 1.** *(Wang et al., 2024c, Lemma 4.3) For two distributions $p, q \in [0, 1]$ over some random variable, denote their respective means by $\bar{p}$ and $\bar{q}$. Then, it holds that*

$$|\bar{p} - \bar{q}| \leq 8\sqrt{\mathrm{Var}(p)D_H^2(p, q)} + 20D_H^2(p, q)$$

*where $\mathrm{Var}(\cdot)$ denotes the variance, and $D_H^2(p, q) := \int (\sqrt{p(x)} - \sqrt{q(x)})^2 \, \mathrm{d}x/2$ is the squared Hellinger distance.*

The following lemma is adapted from some known results from diffusion processes and score matching. It shows that as long as the DS divergence (Definition 1) is small, the Hellinger distance is also small under some mild conditions. Similar results can be found in some theoretical papers on diffusion models such as Chen et al. (2023b); Oko et al. (2023).

**Lemma 2.** *Let $P \in \Delta(\mathbb{R}^d)$ and $g \in \mathbb{R}^d \times [0, T] \to \mathbb{R}^d$. Define $Q \in \Delta(\mathbb{R}^d)$ as the distribution obtained through the reverse process of diffusion starting from $\mathcal{N}(0, I)$ by treating $g$ as the "score function" of the process. Specifically, $Q$ is the distribution of $\mathbf{z}_T$ of the following SDE:*

$$\mathrm{d}\mathbf{z}_t = (\mathbf{z}_t + 2g(\mathbf{z}_t, T - t)) \, \mathrm{d}t + \sqrt{2} \, \mathrm{d}B_t, \quad \mathbf{z}_0 \sim \mathcal{N}(0, I).$$

*We assume the following:*

*(a) For all $t \geq 0$, the score $\nabla \log P_t$ is Lipschitz continuous with finite constant;*

*(b) The second moment of $P$ is upper bounded: $\mathbb{E}_{\mathbf{x} \sim P} \left[ \|\mathbf{x}\|_2^2 \right] \leq m$ for some $m > 0$.*

*Then, if*

$$\underset{\mathbf{x} \sim P}{\mathbb{E}} \underset{t \sim U(T)}{\mathbb{E}} \underset{\mathbf{x}_t \sim q_t(\cdot \mid \mathbf{x})}{\mathbb{E}} \left[ \|g(\mathbf{x}_t, t) - \nabla_{\mathbf{x}_t} \log P_t(\mathbf{x}_t)\|_2^2 \right] \leq \epsilon^2,$$

*for some $\epsilon > 0$, we have*

$$D_H^2(P, Q) = O\left(T\epsilon^2 + (m + d)\exp(-T)\right).$$

*Proof of Lemma 2.* We consider the following three stochastic processes specified by SDEs:

$$
\begin{aligned}
\mathrm{d}\mathbf{x}_t &= (\mathbf{x}_t + 2\nabla \log P_{T-t}(\mathbf{x}_t))\,\mathrm{d}t + \sqrt{2}\,\mathrm{d}B_t,\ \mathbf{x}_0 \sim P_T; \\
\mathrm{d}\mathbf{y}_t &= (\mathbf{y}_t + 2g(\mathbf{y}_t, T-t))\,\mathrm{d}t + \sqrt{2}\,\mathrm{d}B_t,\ \mathbf{y}_0 \sim P_T; \\
\mathrm{d}\mathbf{z}_t &= (\mathbf{z}_t + 2g(\mathbf{z}_t, T-t))\,\mathrm{d}t + \sqrt{2}\,\mathrm{d}B_t,\ \mathbf{z}_0 \sim \mathcal{N}(0, I).
\end{aligned}
$$

To clarify the notation: the variables $\mathbf{x}_t, \mathbf{y}_t, \mathbf{z}_t$ above are defined in the reverse time order, whereas in the lemma statement, $\mathbf{x}$ follow forward time order.

By triangle inequality for Hellinger distance:

$$
\begin{aligned}
D_H^2(P, Q) = D_H^2(\mathbf{x}_T, \mathbf{z}_T) &\leq \left(D_H(\mathbf{x}_T, \mathbf{y}_T) + D_H(\mathbf{y}_T, \mathbf{z}_T)\right)^2 \\
&\leq 2D_H^2(\mathbf{x}_T, \mathbf{y}_T) + 2D_H^2(\mathbf{y}_T, \mathbf{z}_T).
\end{aligned}
$$

We will bound the two terms separately. Since the squared Hellinger distance is upper bounded by KL divergence (i.e., $D_H^2 \leq D_{\mathrm{KL}}$), we seek to establish the KL divergence bounds instead.

First, by Item (a) and Girsanov's Theorem (Karatzas & Shreve, 2014) (also see, e.g., Chen et al. (2023b); Oko et al. (2023)), we have

$$
D_H^2(\mathbf{x}_T, \mathbf{y}_T) \leq D_{\mathrm{KL}}(\mathbf{x}_T \,\|\, \mathbf{y}_T) = O\left(T \cdot \mathbb{E}_{t \sim U(T)} \left\| \nabla \log P_{T-t}(\mathbf{x}_t) - g(\mathbf{x}_t, T-t) \right\|_2^2\right) = O\left(T\epsilon^2\right).
$$

Next, by Item (b) and the convergence of OU process (e.g., Lemma 9 in Chen et al. (2023a)), we have:

$$
D_{\mathrm{KL}}(P_T \,\|\, \mathcal{N}(0, I)) = O\big((m+d) \cdot \exp(-T)\big).
$$

By data-processing inequality for $f$-divergence, we have

$$
\begin{aligned}
D_H^2(\mathbf{y}_T, \mathbf{z}_T) &\leq D_H^2(P_T, \mathcal{N}(0, I)) \\
&\leq D_{\mathrm{KL}}(P_T \,\|\, \mathcal{N}(0, I)) \\
&= O\big((m+d) \cdot \exp(-T)\big).
\end{aligned}
$$

Plugging them back, we complete the proof. $\square$

The following lemmas are standard results from diffusion processes and score matching. We show them here for completeness.

**Lemma 3.** *Given any $g$, we have*

$$
\mathbb{E}_{\mathbf{x}_t \sim p_t} \left\langle g(\mathbf{x}_t, t), \nabla_{\mathbf{x}_t} \log q_t(\mathbf{x}_t) \right\rangle = \mathbb{E}_{\mathbf{x}_0 \sim p_0} \mathbb{E}_{\mathbf{x}_t \sim p_t(\mathbf{x}_t \,|\, \mathbf{x}_0)} \left\langle g(\mathbf{x}_t, t), \nabla_{\mathbf{x}_t} \log q_t(\mathbf{x}_t \,|\, \mathbf{x}_0) \right\rangle
$$

*where $\langle \cdot, \cdot \rangle$ denotes the inner product.*

*Proof of Lemma 3.* It basically follows from integration by parts. We start from the left-hand side (LHS):

$$
\begin{aligned}
\text{LHS} &= \int_{\mathbf{x}_t} g(\mathbf{x}_t, t) \cdot \nabla_{\mathbf{x}_t} \log p_t(\mathbf{x}_t) \cdot p_t(x_t)\,\mathrm{d}\mathbf{x}_t \\
&= \int_{\mathbf{x}_t} g(\mathbf{x}_t, t) \cdot \nabla_{\mathbf{x}_t} p_t(\mathbf{x}_t)\,\mathrm{d}\mathbf{x}_t \\
&= 0 - \int_{\mathbf{x}_t} \nabla_{\mathbf{x}_t} \cdot g(\mathbf{x}_t, t) \cdot p_t(\mathbf{x}_t)\,\mathrm{d}\mathbf{x}_t && \text{(integration by parts)} \\
&= -\int_{\mathbf{x}_0} \int_{\mathbf{x}_t} \nabla_{\mathbf{x}_t} \cdot g(\mathbf{x}_t, t) \cdot p_t(\mathbf{x}_t \,|\, \mathbf{x}_0) \cdot p_0(\mathbf{x}_0)\,\mathrm{d}\mathbf{x}_t\,\mathrm{d}\mathbf{x}_0 \\
&= 0 + \int_{\mathbf{x}_0} \int_{\mathbf{x}_t} g(\mathbf{x}_t, t) \cdot \nabla_{\mathbf{x}_t} p_t(\mathbf{x}_t \,|\, \mathbf{x}_0) \cdot p_0(\mathbf{x}_0)\,\mathrm{d}\mathbf{x}_t\,\mathrm{d}\mathbf{x}_0 && \text{(integration by parts again)} \\
&= \int_{\mathbf{x}_0} \int_{\mathbf{x}_t} g(\mathbf{x}_t, t) \cdot \nabla_{\mathbf{x}_t} \log p_t(\mathbf{x}_t \,|\, \mathbf{x}_0) \cdot p_t(\mathbf{x}_t \,|\, \mathbf{x}_0) \cdot p_0(\mathbf{x}_0)\,\mathrm{d}\mathbf{x}_t\,\mathrm{d}\mathbf{x}_0 \\
&= \text{RHS}.
\end{aligned}
$$

Hence, the lemma is proved. $\square$

**Lemma 4.** *Given any $g$, we have*

$$\mathop{\mathbb{E}}_{\mathbf{x}_t \sim p_t} \left\| g(\mathbf{x}_t, t) - \nabla_{\mathbf{x}_t} \log p_t(\mathbf{x}_t) \right\|_2^2 - \mathop{\mathbb{E}}_{\mathbf{x}_0 \sim p_0, \mathbf{x}_t \sim p_t(\mathbf{x}_t \mid \mathbf{x}_0)} \left\| g(\mathbf{x}_t, t) - \nabla_{\mathbf{x}_t} \log p_t(\mathbf{x}_t \mid \mathbf{x}_0) \right\|_2^2$$

$$= \mathop{\mathbb{E}}_{\mathbf{x}_t} \left\| \nabla_{\mathbf{x}_t} \log p_t(\mathbf{x}_t) \right\|_2^2 - \mathop{\mathbb{E}}_{\mathbf{x}_0, \mathbf{x}_t} \left\| \nabla_{\mathbf{x}_t} \log p_t(\mathbf{x}_t \mid \mathbf{x}_0) \right\|_2^2$$

*where we note that the right-hand side of the equation is independent of $g$.*

*Proof of Lemma 4.* We prove the lemma by expanding the first term on the left-hand side:

$$\mathop{\mathbb{E}}_{\mathbf{x}_t \sim p_t} \left\| g(\mathbf{x}_t, t) - \nabla_{\mathbf{x}_t} \log p_t(\mathbf{x}_t) \right\|_2^2$$

$$= \mathop{\mathbb{E}}_{\mathbf{x}_t \sim p_t} \left\| g(\mathbf{x}_t, t) \right\|_2^2 - 2 \mathop{\mathbb{E}}_{\mathbf{x}_t \sim p_t} \left\langle g(\mathbf{x}_t, t), \nabla_{\mathbf{x}_t} \log p_t(\mathbf{x}_t) \right\rangle + \mathop{\mathbb{E}}_{\mathbf{x}_t \sim p_t} \left\| \nabla_{\mathbf{x}_t} \log p_t(\mathbf{x}_t) \right\|_2^2$$

$$= \mathop{\mathbb{E}}_{\mathbf{x}_0, \mathbf{x}_t} \left\| g(\mathbf{x}_t, t) \right\|_2^2 - 2 \mathop{\mathbb{E}}_{\mathbf{x}_0, \mathbf{x}_t} \left\langle g(\mathbf{x}_t, t), \nabla_{\mathbf{x}_t} \log p_t(\mathbf{x}_t \mid \mathbf{x}_0) \right\rangle + \mathop{\mathbb{E}}_{\mathbf{x}_t} \left\| \nabla_{\mathbf{x}_t} \log p_t(\mathbf{x}_t) \right\|_2^2 \quad \text{(Lemma 3)}$$

$$= \mathop{\mathbb{E}}_{\mathbf{x}_0, \mathbf{x}_t} \left\| g(\mathbf{x}_t, t) \right\|_2^2 - 2 \mathop{\mathbb{E}}_{\mathbf{x}_0, \mathbf{x}_t} \left\langle g(\mathbf{x}_t, t), \nabla_{\mathbf{x}_t} \log p_t(\mathbf{x}_t \mid \mathbf{x}_0) \right\rangle + \mathop{\mathbb{E}}_{\mathbf{x}_0, \mathbf{x}_t} \left\| \nabla_{\mathbf{x}_t} \log p_t(\mathbf{x}_t \mid \mathbf{x}_0) \right\|_2^2$$

$$- \mathop{\mathbb{E}}_{\mathbf{x}_0, \mathbf{x}_t} \left\| \nabla_{\mathbf{x}_t} \log p_t(\mathbf{x}_t \mid \mathbf{x}_0) \right\|_2^2 + \mathop{\mathbb{E}}_{\mathbf{x}_t} \left\| \nabla_{\mathbf{x}_t} \log p_t(\mathbf{x}_t) \right\|_2^2$$

$$= \mathop{\mathbb{E}}_{\mathbf{x}_0, \mathbf{x}_t} \left\| g(\mathbf{x}_t, t) - \nabla_{\mathbf{x}_t} \log p_t(\mathbf{x}_t \mid \mathbf{x}_0) \right\|_2^2 - \mathop{\mathbb{E}}_{\mathbf{x}_0, \mathbf{x}_t} \left\| \nabla_{\mathbf{x}_t} \log p_t(\mathbf{x}_t \mid \mathbf{x}_0) \right\|_2^2 + \mathop{\mathbb{E}}_{\mathbf{x}_t} \left\| \nabla_{\mathbf{x}_t} \log p_t(\mathbf{x}_t) \right\|_2^2.$$

This completes the proof. $\square$

### C.2 STATISTICAL RESULTS

For the ease of presentation, we will leverage the following quantities:

$$\mathcal{L}(\pi, g) = \mathop{\mathbb{E}}_{s \sim d^\pi} \mathop{\mathbb{E}}_{t \sim U(T)} \mathop{\mathbb{E}}_{s_t \sim q_t(\cdot \mid s)} \left[ \|g^e(s_t, t) - \nabla_{s_t} \log q_t(s_t \mid s)\|_2^2 - \|g(s_t, t) - \nabla_{s_t} \log q_t(s_t \mid s)\|_2^2 \right]$$

By Lemma 4, the above is equivalent to the following by replacing the conditional score functions with the marginal ones for both terms in the expectation:

$$\mathcal{L}(\pi, g) = \mathop{\mathbb{E}}_{s \sim d^\pi} \mathop{\mathbb{E}}_{t \sim U(T)} \mathop{\mathbb{E}}_{s_t \sim q_t(\cdot \mid s)} \left[ \|g^e(s_t, t) - \nabla_{s_t} \log p_t^\pi(s_t)\|_2^2 - \|g(s_t, t) - \nabla_{s_t} \log p_t^\pi(s_t)\|_2^2 \right]. \tag{4}$$

One can show that, fixing a policy $\pi$, we have

$$\max_g \mathcal{L}(\pi, g) = \mathop{\mathbb{E}}_{s \sim d^\pi} \mathop{\mathbb{E}}_{t \sim U(T)} \mathop{\mathbb{E}}_{s_t \sim q_t(\cdot \mid s)} \left[ \|g^e(s_t, t) - \nabla_{s_t} \log p_t^\pi(s_t)\|_2^2 \right] = \ell(\pi) \tag{5}$$

where the first equality is by observing that the second term in Eq. (4) can be minimized to zero by choosing $g = \nabla_{s_t} \log p_t^\pi(s_t)$ (recalling that we do not assume misspecification for score functions), and the second equality is by definition.

Bounding the difference between $\pi^{(1:K)}$ and $\pi^e$ is what we aim to do in the following lemma.

**Lemma 5.** *Under Assumptions 1 and 2, assuming misspecification of RL (i.e., $\pi^e$ may not be in $\Pi$), we have*

$$D_H^2(d^{\pi^{(1:K)}}, d^{\pi^e}) = O\left( T \cdot \epsilon_{\text{score}}^2 + T \cdot \epsilon_{\text{mis}} + T \cdot \epsilon_{\text{RL}} + \frac{T \cdot \text{Regret}(K)}{K} + (m + d) \exp(-T) \right).$$

*Setting $T = \log(\frac{m+d}{\epsilon_{\text{score}}^2 + \epsilon_{\text{RL}} + \epsilon_{\text{mis}}})$ yields the following bound:*

$$D_H^2(d^{\pi^{(1:K)}}, d^{\pi^e}) = \widetilde{O}\left( \epsilon_{\text{score}}^2 + \epsilon_{\text{mis}} + \epsilon_{\text{RL}} + \frac{\text{Regret}(K)}{K} \right)$$

*where $\widetilde{O}$ hides logarithmic factors.*

*Proof of Lemma 5.* Items (b) and (c) in Assumption 1 implies the following:

$$\max_g \sum_{k=1}^K \mathcal{L}(\pi^{(k)}, g) \le \sum_{k=1}^K \mathcal{L}(\pi^{(k)}, g^{(k)}) + \text{Regret}(K); \tag{6}$$

$$\mathcal{L}(\pi^{(k)}, g^{(k)}) - \min_{\pi \in \Pi} \mathcal{L}(\pi, g^{(k)}) \le \epsilon_{\text{RL}}. \tag{7}$$

Combining Equation (6) and Equation (7), we have

$$\max_g \sum_{k=1}^K \mathcal{L}(\pi^{(k)}, g) \le \sum_{k=1}^K \min_{\pi \in \Pi} \mathcal{L}(\pi, g^{(k)}) + K \cdot \epsilon_{\text{RL}} + \text{Regret}(K).$$

Now the right-hand side can be upper bounded by replacing all $g^{(k)}$ with maximization over $g$:

$$\max_g \sum_{k=1}^K \mathcal{L}(\pi^{(k)}, g) \le \sum_{k=1}^K \min_{\pi \in \Pi} \max_g \mathcal{L}(\pi, g) + K \cdot \epsilon_{\text{RL}} + \text{Regret}(K)$$

$$= \sum_{k=1}^K \min_{\pi \in \Pi} \ell(\pi) + K \cdot \epsilon_{\text{RL}} + \text{Regret}(K) \qquad \text{(by Eq. 5)}$$

$$= K \cdot \epsilon_{\text{mis}} + K \cdot \epsilon_{\text{RL}} + \text{Regret}(K) \tag{8}$$

Note that the left-hand side is equivalent to the following

$$\max_g \sum_{k=1}^K \mathcal{L}(\pi^{(k)}, g) = K \cdot \max_g \mathcal{L}(\pi^{(1:K)}, g) = K \cdot \ell(\pi^{(1:K)}).$$

where the first equality is by the definition of $\mathcal{L}$. Inserting it back to the previous inequality yields an upper bound on $\ell(\pi^{(1:K)})$:

$$\ell(\pi^{(1:K)}) \le \epsilon_{\text{mis}} + \epsilon_{\text{RL}} + \frac{\text{Regret}(K)}{K}$$

Now we define a new distribution to facilitate the analysis: let $\widehat{d^{\pi^e}}$ denote the distribution induced by the reverse diffusion process starting from $\mathcal{N}(0, I)$ by treating $g^e$ as the "score function". Specifically, $\widehat{d^{\pi^e}}$ is the distribution of $\overline{s}_T$ of the following SDE:

$$d\overline{s}_t = (\overline{s}_t + 2g^e(\overline{s}_t, T - t)) \, dt + \sqrt{2} \, dB_t, \quad \overline{s}_0 \sim \mathcal{N}(0, I)$$

where we recall that $B_t$ is the standard Brownian motion. We can see that $\widehat{d^{\pi^e}}$ is approximating $d^{\pi^e}$. Then, we invoke Lemma 2 and get

$$D_H^2(d^{\pi^{(1:K)}}, \widehat{d^{\pi^e}}) = O\left(T \cdot \epsilon_{\text{mis}} + T \cdot \epsilon_{\text{RL}} + \frac{T \cdot \text{Regret}(K)}{K} + (m + d)\exp(-T)\right).$$

Now it remains to bound $D_H^2(d^{\pi^e}, \widehat{d^{\pi^e}})$ so that we can apply the triangle inequality to get an upper bound on $D_H^2(d^{\pi^{(1:K)}}, d^{\pi^e})$. Since $\ell(\pi^e) \le \epsilon_{\text{score}}^2$ (Item (a) in Assumption 1), we invoke Lemma 2 again and immediately get

$$D_H^2(d^{\pi^e}, \widehat{d^{\pi^e}}) \le O\left(T \cdot \epsilon_{\text{score}}^2 + (m + d)\exp(-T)\right).$$

We conclude the proof by putting them together via triangle inequality:

$$D_H^2(d^{\pi^{(1:K)}}, d^{\pi^e}) \le 2D_H^2(d^{\pi^{(1:K)}}, \widehat{d^{\pi^e}}) + 2D_H^2(\widehat{d^{\pi^e}}, d^{\pi^e})$$

$$= O\left(T \cdot \epsilon_{\text{mis}} + T \cdot \epsilon_{\text{score}}^2 + T \cdot \epsilon_{\text{RL}} + \frac{T \cdot \text{Regret}(K)}{K} + (m + d)\exp(-T)\right).$$

$\square$

**Lemma 6.** *Under the same condition as Lemma 5 but assuming no misspecification (i.e., $\pi^e \in \Pi$), we have*

$$D_H^2(d^{\pi^{(1:K)}}, d^{\pi^e}) = \widetilde{O}\left(\epsilon_{\text{score}}^2 + \epsilon_{\text{RL}} + \frac{\text{Regret}(K)}{K}\right)$$

*Proof of Lemma 6.* The proof is almost identical to that of Lemma 5 except that, when there is no misspecification, $\min_{\pi \in \Pi} \ell(\pi)$ in Eq. 8 can be simply upper bounded by $\ell(\pi^e)$, which is bounded by $\epsilon_{\text{score}}^2$ (Item (a) in Assumption 1). The rest of the proof is the same. $\square$

### C.3 Main proof

In this section, we provide the main proof of Theorem 1. First, the following is by definition of value functions:

$$V^{\pi^{(1:K)}} - V^{\pi^e} = H \cdot \left( \mathop{\mathbb{E}}_{s,a \sim d^{\pi^{(1:K)}}} c^\star(s,a) - \mathop{\mathbb{E}}_{s,a \sim d^{\pi^e}} c^\star(s,a) \right)$$

By Lemma 1, it is bounded by

$$V^{\pi^{(1:K)}} - V^{\pi^e} \tag{9}$$

$$\leq H \cdot \left( 8 \sqrt{\mathrm{Var}\left( \mathop{\mathbb{E}}_{s,a \sim d^{\pi^e}} c^\star(s,a) \right) \cdot D_H^2(c^\star \sim d^{\pi^{(1:K)}}, c^\star \sim d^{\pi^e}) + 20 D_H^2(c^\star \sim d^{\pi^{(1:K)}}, c^\star \sim d^{\pi^e})} \right)$$

$$= 8 \sqrt{\mathrm{Var}^{\pi^e} \cdot D_H^2(c^\star \sim d^{\pi^{(1:K)}}, c^\star \sim d^{\pi^e})} + 20 H D_H^2(c^\star \sim d^{\pi^{(1:K)}}, c^\star \sim d^{\pi^e})$$

where $c^\star \sim d^{\pi^{(1:K)}}$ (and $c^\star \sim d^{\pi^e}$) denotes the distribution of cost rather than the distribution of $d^{\pi^{(1:K)}}$ (and $d^{\pi^e}$) itself. Now we apply the data processing inequality and get

$$V^{\pi^{(1:K)}} - V^{\pi^e} \leq 8 \sqrt{\mathrm{Var}^{\pi^e} \cdot D_H^2(d^{\pi^{(1:K)}}, d^{\pi^e})} + 20 H D_H^2(d^{\pi^{(1:K)}}, d^{\pi^e}).$$

Plugging in the bound from Lemma 5 if there is misspecification error or Lemma 6 otherwise, we get

$$V^{\pi^{(1:K)}} - V^{\pi^e} = \widetilde{O}\left( \sqrt{\mathrm{Var}^{\pi^e} \cdot \epsilon} + \epsilon H \right).$$

Similarly, we can derive the second half of the second-order bound (with $V^{\pi^{(1:K)}}$ replaced with $V^{\pi^e}$ inside the square root) by applying Lemma 1 to Eq. 9 in the other direction. These conclude the proof of the second-order bound.

The first-order bound is a direct consequence of the second-order bound as shown in Theorem 2.1 in Wang et al. (2023).

## D Additional Experimental Results

In this section, we present additional experimental results: in Appendix D.1, we show the results of learning from state-action demonstrations; in Appendix D.2, we analyze the impact of the number of expert demonstrations on performance; in Appendix D.3, we study the expressiveness of diffusion models over other methods; in Appendix D.4, we include plots of unnormalized returns for reference; finally, in Appendix D.5, we provide detailed implementation details.

In addition, visual demonstrations of the experiments (such as time-lapse images) are provided in Figure 3 for better illustration.

### D.1 Learning from State-action Demonstrations

Our method can be easily extended to learning from state-action demonstrations by appending the action vector to the state vector for both training and computing rewards. Hence, we also explore its performance in such a setting. The experimental setup is identical to that of learning from state-only data. The training curves are presented in Figure 4. The results are highly consistent with those from the state-only setting, indicating that the performance gap is stable whether training on state data alone or state-action pairs. This also demonstrates that SMILING is robust to different data types.

### D.2 On the Number of Expert Demonstrations

As we established in Section 5, our algorithm can have better sample complexity in terms of expert demonstrations. To empirically verify this, we conduct an experiment with varying numbers of

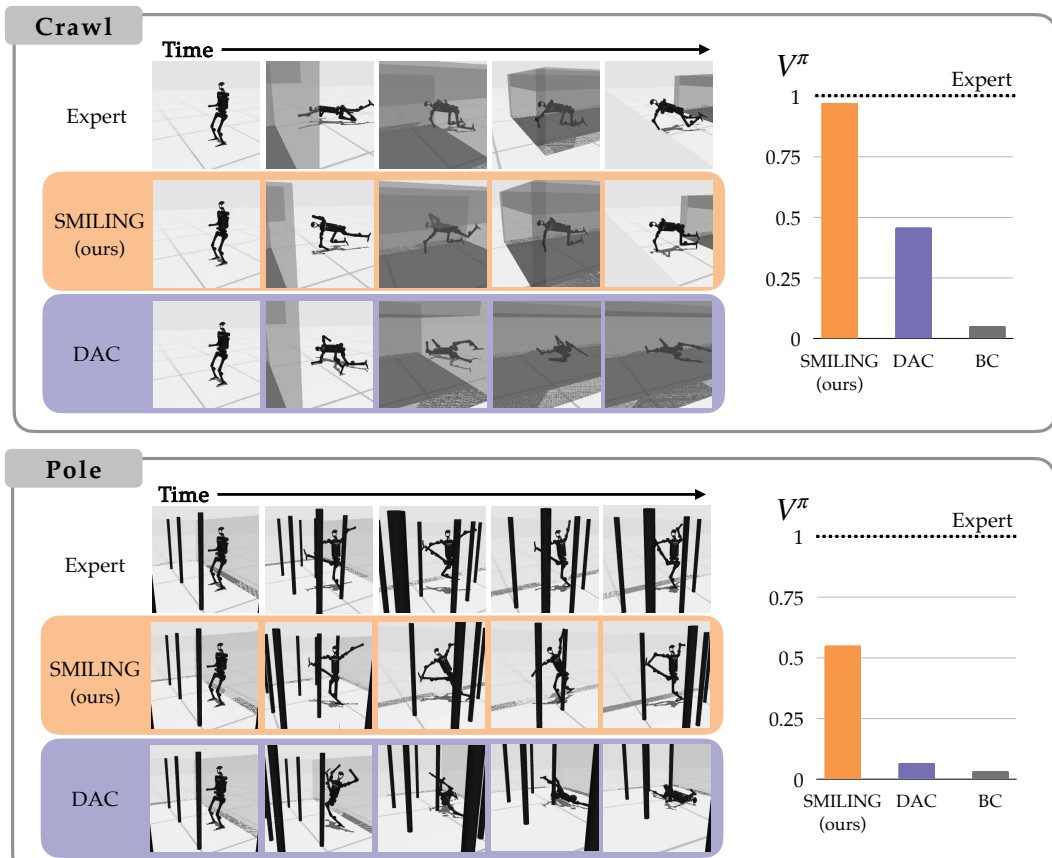

Figure 3: **Whole-body humanoid control via IL from state alone**. The two panels illustrate the `crawl` and `pole` tasks, respectively. In both tasks, we show the time-lapse frames of the expert policy and the policies learned by our method (`SMILING`) and DAC after 3M training steps. In the `crawl` task, the goal is to crawl through a grey tunnel, where both the expert and ours succeed and the crawling movements are similar. However, DAC collapses and fails to complete the task. In the `pole` task, the goal is to travel through a dense forest of poles. Ours successfully navigates through the poles, though with less stability than the expert, while DAC collapses to the ground and cannot move. The bar graphs on the right show normalized policy performance, where `SMILING` significantly outperforms DAC and Behavioral Cloning (BC) in both tasks, approaching expert performance in `crawl`. Note that BC uses expert actions, while DAC and `SMILING` learn from states alone.

expert demonstrations. We train `SMILING` and DAC with 1K, 2K, 5K, 10K, 25K, and 125K expert states on `humanoid-walk` and compare the results. The relationship between the number of expert demonstrations and the final performance (after 3M training steps) is shown in Figure 5 and the training curves of both algorithms are shown in Figure 6. From Figure 5, we observe that `SMILING` dominates DAC across all expert demonstration sizes. In particular, `SMILING` is able to approach expert-level performance with 5K expert states, while DAC requires 25K to achieve similar performance. This suggests that `SMILING` is more efficient than DAC in utilizing expert demonstrations.

### D.3 EXPRESSIVENESS OF DIFFUSION MODELS

In Appendix A, we argued that score matching is more expressive than discriminator-based methods, which may explain why our approach generally outperforms DAC in previous experiments. To further support this argument, we conducted an ablation study on the `cheetah-run` task where our method previously showed a slower convergence compared to DAC. Now we remove the activation functions in both the discriminator of DAC and the score function of our method, making

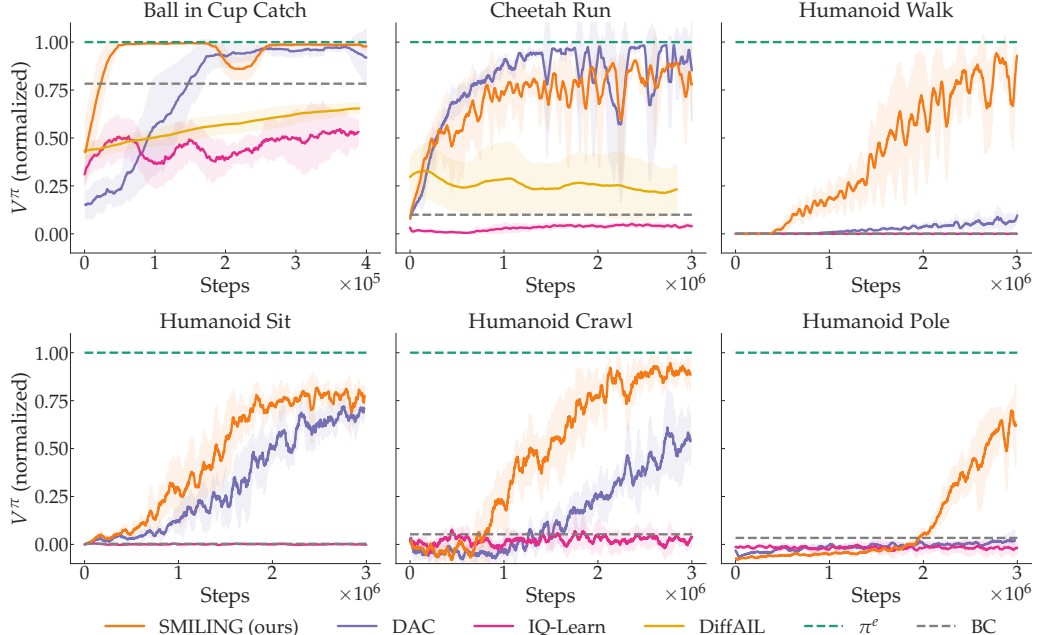

Figure 4: Learning curves for learning from state-action data across five random seeds. The x-axis corresponds to the number of environment steps (also the number of policy updates). The y-axis is normalized such that the expert performance is one and the random policy is zero. The results are consistent with those from the state-only setting.

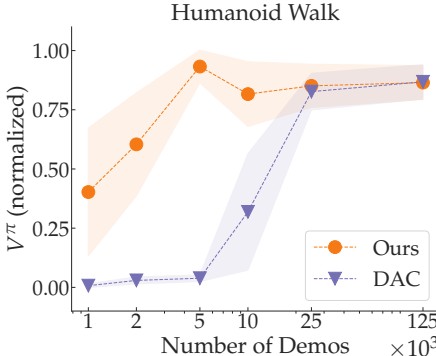

Figure 5: Relationship between the number of expert demonstrations and the final performance. The x-axis corresponds to the number of expert states. The y-axis is the final performance that is computed as the average of the last 100 training epochs. Each point is the average of five random seeds.

them *purely linear*, and then re-run the experiments. The results are in Figure 7. Different from the previous experiments where DAC outperformed our method, in this case, DAC's performance degrades notably by showing slower convergence and significantly higher variance. In comparison, our method remains more effective and outperforms DAC clearly.

We attribute the performance degradation of DAC to the limited expressiveness of the discriminator, which results in mode collapse—a well-known issue in discriminator-based methods, originally identified in GANs. In such methods, if the discriminator's performance falls significantly behind the generator (e.g., due to insufficient training or a lack of discriminator expressiveness, which is the case here), the generator will exploit this gap and produce single-mode data that can easily fool the discriminator. In this case, the generator is falling into a local minima. As the discriminator gradually catches up and learns to identify this data, the generator will discover new data to fool the updated discriminator, falling into another local minimum. This perpetual cycle of chasing

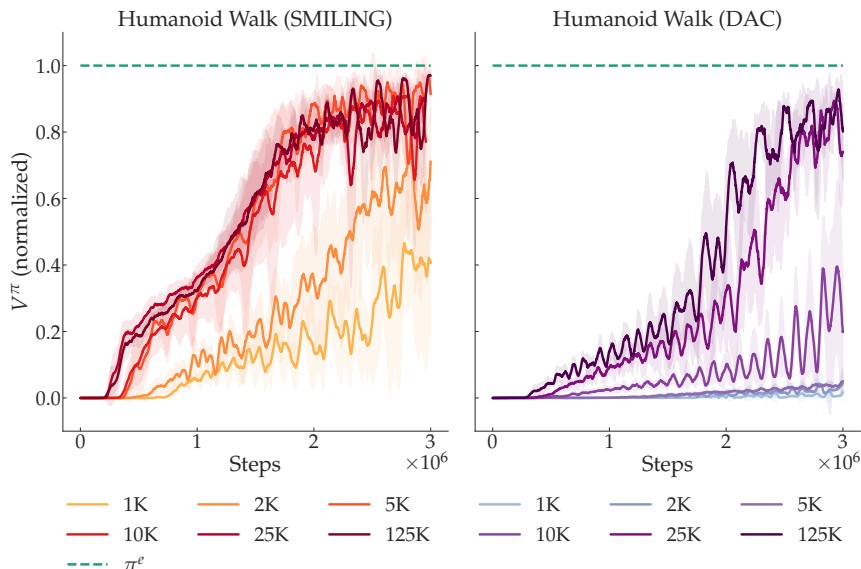

Figure 6: The training curves of SMILING (left) and DAC (right) with varying numbers of expert demonstrations. The x-axis corresponds to the number of environment steps (also the number of policy updates). The y-axis is the normalized return.

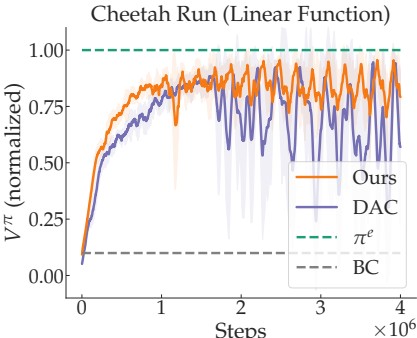

Figure 7: Learning curves with linear functions. Compared to using MLPs, DAC shows a significant decline in performance of slower convergence and higher variance, while ours remains more effective.

leads to continuous oscillations in the training process. We conjecture this is the exact reason for DAC's instability in this experiments—With the discriminator reduced to a purely linear function, its expressiveness is significantly weakened, allowing the RL policy to exploit it easily, and thus a perpetual chasing emerges. However, our method can remain more effective even with linear score functions, which aligns with our hypothesis in Appendix A.

### D.4 PLOTS WITH UNNORMALIZED RETURNS

We provide the unnormalized-return versions (i.e., with the y-axis not normalized) of Figures 2 and 4 in Figures 8 and 9 for reference.

### D.5 IMPLEMENTATION DETAILS

We summarize the difficulties of the tasks evaluated in our experiments in Table 1.

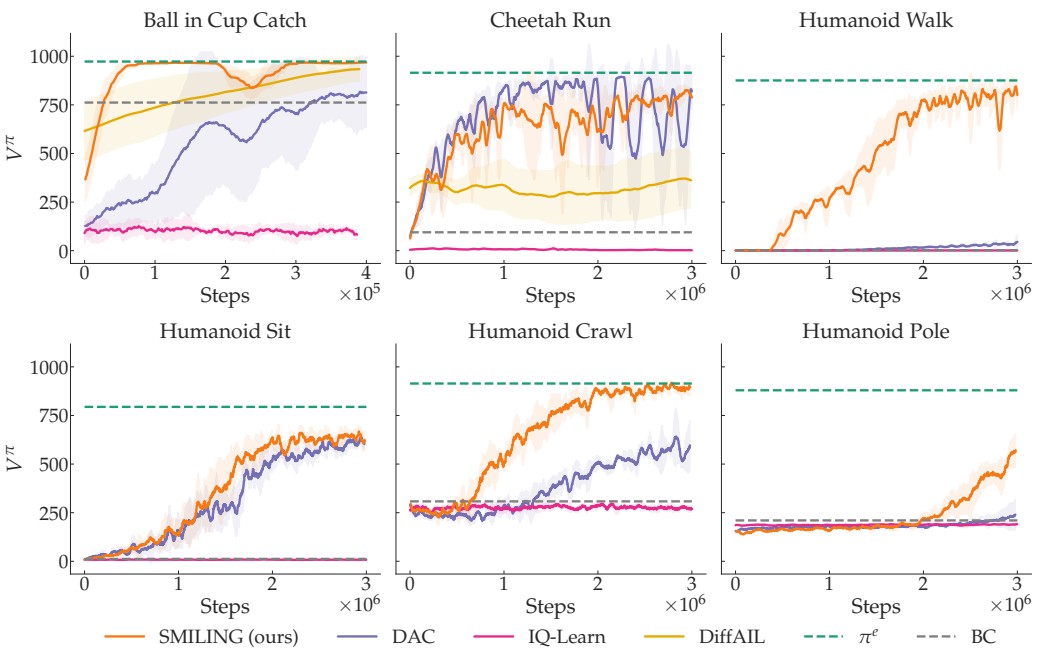

Figure 8: Same curves as in Figure 2 but with unnormalized return $V^\pi$ on the y-axis.

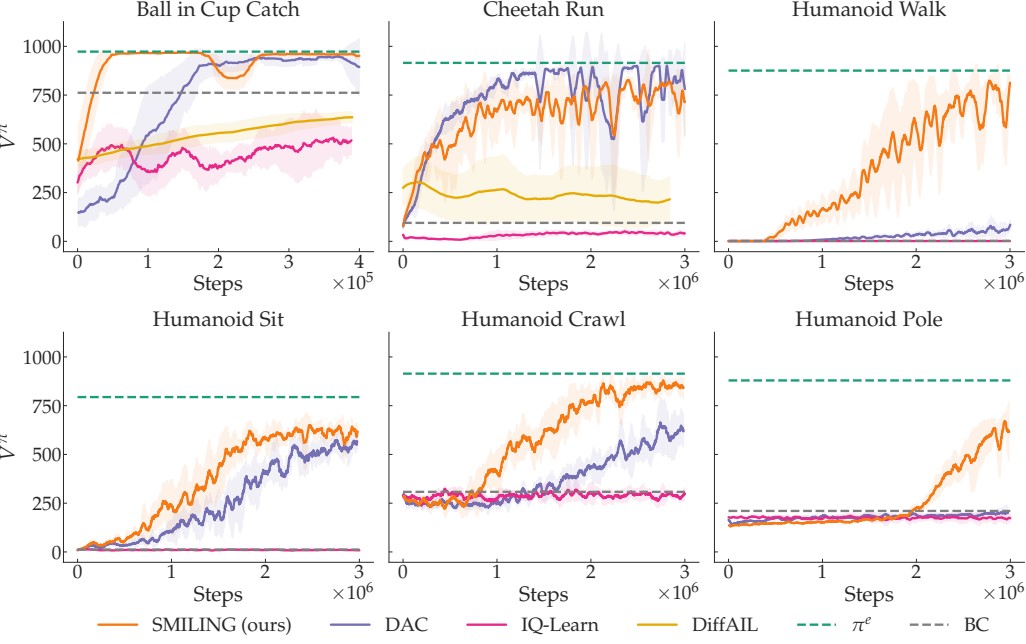

Figure 9: Same curves as in Figure 4 but with unnormalized return $V^\pi$ on the y-axis.

**Implementation of RL Algorithms.** We use the SAC implementation from Yarats & Kostrikov (2020) and the DreamerV3 implementation from Sferrazza et al. (2024) without modifying their hyperparameters. The only change we made is substituting their ground-truth cost (reward) functions with our cost function. For example, whenever the SAC agent samples a batch of transitions from

the replay buffer, we replace the corresponding cost values with those defined in our algorithm. We do the same for DreamerV3.

**Expert Training and Demonstration Collection.** We use the aforementioned RL algorithms (SAC for DMC tasks and DreamerV3 for HumanoidBench tasks) to train the expert policies. We did not change the hyperparameters for these algorithms, making them the same as those provided in their original implementations in Yarats & Kostrikov (2020); Sferrazza et al. (2024). In other words, these algorithms were directly applied to their respective benchmarks without modification.

The RL algorithms were run until the performance plateaued, with the required number of training steps varying across tasks: `ball-in-cup-catch` (0.1M steps), `cheetah-run` (4.8M steps), `humanoid-walk` (3.4M steps), `humanoid-sit` (2.7M steps), `humanoid-crawl` (5.4M steps), and `humanoid-pole` (16M steps). The resulting expert policies achieve the following average total return: 972.85 for `ball-in-cup-catch`, 915.16 for `cheetah-run`, 875.61 for `humanoid-walk`, 794.12 for `humanoid-sit`, 914.31 for `humanoid-crawl`, and 879.48 for `humanoid-pole`.

Once the expert training is complete, we generate expert demonstrations by running the learned expert policy for five episodes for each task. Since each episode has a fixed horizon of 1K steps, the resulting dataset contains $5 \times 1,000 = 5,000$ samples per task.

**Implementation of Our Method and DAC.** We made efforts to align the implementation details of both algorithms to ensure a fair comparison. A comprehensive list of hyperparameters is provided in Table 2. We perform one update of the score function in our method and the discriminator in DAC per 1K RL steps. The expert diffusion model is trained for 4K epochs using 5K expert states (and actions, if learning from state-action pairs).

**Implementation of IQ-Learn.** We used the official implementation from Garg et al. (2021). We note that, unlike `SMILING` and DAC that can be built on any RL algorithm, IQ-Learn is exclusively tied to SAC. This makes it unclear how to implement it on top of DreamerV3, as we did for `SMILING` and DAC. Therefore, we adhered to their SAC implementation to run all experiments including the HumanoidBench tasks.

**Implementation of DiffAIL.** We used the official implementation provided by Wang et al. (2024a) and made sure their neural networks have same size as ours. However, when applying their code to the tasks involving humanoids, we encountered numerical issues: the loss and weights of the neural networks diverged to infinity. We suspect this is due to numerical instability in their implementation, which becomes pronounced in the high-dimensional feature space of humanoid tasks.

Table 1: List of environments ("DMC" = DeepMind Control Suite, "HB" = HumanoidBench).

| Task | Difficulty | Benchmark |
|---|---|---|
| Ball in Cup Catch | Easy | DMC |
| Cheetah Run | Medium | DMC |
| Humanoid Walk | Hard | DMC |
| Humanoid Sit | Hard | HB |
| Humanoid Crawl | Hard | HB |
| Humanoid Pole | Hard | HB |

Table 2: Hyperparameters of our method and DAC. Shared columns indicate same hyperparameters.

| Hyperparameter | Ours | DAC |
|---|---|---|
| Neural Net | MLP (1 hidden layer w/ 256 units) | |
| Learning rate | $5 \times 10^{-3}$ | |
| Samples used per update | 100,000 | |
| Batch size | 1,024 | |
| Discretization steps | 5,000 | N/A |
| Time embedding size | 16 | N/A |

