# OpenReview forum: "Diffusing States and Matching Scores: A New Framework for Imitation Learning"
_ICLR.cc/2025/Conference — ICLR 2025 Poster_

### Official Review · Reviewer_H2TW · 2024-10-28

**Soundness:** 3
**Presentation:** 3
**Contribution:** 3
**Rating:** 8
**Confidence:** 3

**Summary:**

This paper introduces an Inverse Reinforcement Learning (IRL) algorithm called SMILING, which replaces the commonly used f-divergence with a novel approach called Diffusion Score Divergence. This new method measures the discrepancy between the expert's and the learner's state distributions along the same forward diffusion process. The proposed algorithm achieves significantly tighter first- and second-order instance-dependent bounds, supported by thorough error analysis throughout the entire algorithm. Experimental results demonstrate the effectiveness of SMILING across various standard control benchmarks.

**Strengths:**

I appreciate the numerous contributions and the excellent presentation of this paper:

- The idea is an innovative application of score matching techniques to model the state distributions of the policy. Unlike previous approaches that rely on training a discriminator between expert and learner data, the theoretical results demonstrate that the score-based method is more effective and sample-efficient, avoiding issues such as unstable training and mode collapse.
- The presentation is clear and highly informative. I particularly enjoyed the writing in Section 5.1, which intuitively explains why the score-based approach is more expressive than discriminators, especially in the context of linear models. The analysis in Section 5.1 is further validated by the experiments in Section 6.3, which is a great reinforcement of the theoretical insights.
- Across various benchmarks, the method provides a significant improvement over baseline techniques in the experiments, showing strong empirical results.

Overall, the paper's theoretical and experimental results convincingly demonstrate the novelty and effectiveness of the proposed method. This approach could potentially inspire other fields that rely on training discriminators to explore score-based techniques.

**Weaknesses:**

Although this paper presents promising results, there are three primary weaknesses that should be addressed to some extent. I will consider any responses to these concerns when recalibrating my final score.

- **Time Complexity of Evaluating the Cost Function $c^k(s)$ in Eq.3**:
  - The main concern is the time complexity associated with evaluating the cost function. In the experimental sections, the authors employ a DDPM framework with over 5,000 steps. This implies that evaluating the cost function for a given state requires 5,000 evaluations of the model, which could be prohibitive for more complex scenarios, particularly with higher-dimensional state spaces.
  - If the authors are using a single randomly sampled time step to estimate Eq.3, the variance of this estimation across time should be incorporated into the final theoretical results. However, this issue is not addressed in the current analysis, raising questions about the method's scalability to high-dimensional state spaces.
- **Choice of Data Aggregation**:
  - In Line 3 of the algorithm, the score matching target aims to force $g(s_t, t)$ to model the marginal score of the state distributions from a mixture of policies $\pi^{(1:k-1)}$. However, the paper lacks a clear justification for using this mixture-based approach instead of the more intuitive choice of selecting the prior learned policy $\pi^{k-1}$. While the theoretical results focus on the mixture of policies, it would be valuable to demonstrate the necessity of this choice.
  - Providing more empirical or theoretical results to verify the effectiveness of this data aggregation technique during training would help clarify its benefits. For instance, comparing the proposed approach with a baseline that only uses $\pi^{k-1}$ could offer insights into the significance of data aggregation.
- **Assumption (c) and the Rate of $\epsilon_{RL}$ in Continuous State-Space Settings**:
  - The rate of $\epsilon_{RL}$ should be discussed in more detail, especially considering that score matching is typically applied in continuous spaces. Although some recent algorithms extend score matching to discrete spaces [1, 2], incorporating them into the current framework may not be straightforward.
  - It would be beneficial to discuss how the proposed framework might be adapted or fit into established theoretical results in discrete spaces. This could include exploring potential modifications or providing insights into the limitations of the current approach when dealing with discrete settings.

Addressing these concerns would significantly strengthen the paper by enhancing its scalability, clarifying the choice of data aggregation, and broadening the discussion on the applicability of the proposed framework in various state-space settings.

[1] Meng, Chenlin, et al. "Concrete score matching: Generalized score matching for discrete data." *Advances in Neural Information Processing Systems* 35 (2022): 34532-34545.

[2] Lou, Aaron, Chenlin Meng, and Stefano  Ermon. "Discrete diffusion language modeling by estimating the ratios of the data distribution." *arXiv preprint arXiv:2310.16834* (2023).

**Questions:**

The same with the weaknesses I mentioned above.

---

> ### Author Response · Authors · 2024-11-23
>
> Thank you for your valuable feedback! Your questions are insightful, and we would like to share our thoughts on them below.
>
> - Time Complexity of Evaluating the Cost Function $c^k(s)$ in Eq.3:
>
> Our thoughts on this are twofold:
>
> (1) Estimating the cost function does not require 5000 steps to be executed sequentially (i.e., 5000 steps one by one). This process is fully parallelized. In practice, we use pytorch’s parallelization techniques to optimize this step efficiently during our experiments.
>
> (2) In fact, we don’t need to sample 5000 points to estimate the cost. Concentration inequalities imply that the estimation error decreases rapidly with the number of samples. In our experiments, we found that approximately 100–500 samples are sufficient.
>
> Moreover, even if our algorithm may consume more computing resource, this is a training time only cost--it does not mean the learned policy is slow in running. We can also draw an analogy to image generation: while GANs generate images in a single pass, diffusion models require multiple steps; despite being slower, diffusion models produce higher-quality images.
>
>
> - Choice of Data Aggregation: why learning from the mixture of $\pi^{(1:k-1)}$ rather than $\pi^{k-1}$
>
> Empirically, learning from mixture policy is more stable and allows us to use a larger dataset for training, which helps with finite sample concerns. Additionally, the key reason why DAC outperforms GAIL is that DAC learns from a mixture of past policies, while GAIL learns solely on the most recent policy. This point is empirically observable and is also argued in the DAC paper.
>
>
> - The rate of $\epsilon_\text{RL}$
>
> The rate of $\epsilon_\text{RL}$ is a core question in RL theory. Numerous studies have explored and established rates for $\epsilon_\text{RL}$ under various settings, including those with continuous space--to list a few, we have the following:
>
> [1] show that, for linear MDP, the rate is upper bounded by $O(\sqrt{d^3 H/M})$ where $d$ is the feature dimension and $M$ is the number of rollout trajectories. For linear mixture MDP, we have rate of $O(d/\sqrt{M})$ from [2]. More generally, [3] establishes the rate for low-rank MDPs, while [4] extends this to low-nonnegative-rank MDPs. Recall that we defined $\epsilon_\text{RL}$ as the upper bound on returns averaged over all $h$ so the $H$ dependence above is decreased.
>
> The question of the rate of $\epsilon_\text{RL}$ is orthogonal to this work. In fact, any improvement in the rate of $\epsilon_\text{RL}$ can be seamlessly integrated into the theoretical framework we present.
>
> [1] Jin, Chi, et al. "Provably efficient reinforcement learning with linear function approximation." Mathematics of Operations Research 48.3 (2023): 1496-1521.
>
> [2] Zhou, Dongruo, Quanquan Gu, and Csaba Szepesvari. "Nearly minimax optimal reinforcement learning for linear mixture markov decision processes." Conference on Learning Theory. PMLR, 2021.
>
> [3] Agarwal, Alekh, et al. "Flambe: Structural complexity and representation learning of low rank mdps." Advances in neural information processing systems 33 (2020): 20095-20107.
>
> [4] Modi, Aditya, et al. "Model-free representation learning and exploration in low-rank mdps." Journal of Machine Learning Research 25.6 (2024): 1-76.
>
> - how the proposed framework might be adapted or fit into established theoretical results in discrete spaces
>
> Thanks for bringing this interesting point! In this work, we focus on continuous robotics control which itself is an important problem. Regarding discrete one, we have the following thought:
>
> We have treated the diffusion model as a black box in this work. All we require is its training loss (score matching loss) to compute reward signal. We are entirely agnostic to its internal structure. Therefore, for discrete spaces, any new score-matching technique can be directly integrated into our framework and we can apply our algorithm in the same way (by just changing the score matching component in the algo).
>
> Hence, we believe this is highly feasible. In this sense, our algorithm could be extensible and adaptable to future developments like this.

---

> > ### Comment · Reviewer_H2TW · 2024-11-24
> > **Raise Score**
> >
> > I thank the authors for their detailed answers, which address most of my concerns and I have raised my score to 8. Good luck!

---

### Official Review · Reviewer_yQfZ · 2024-11-03

**Soundness:** 4
**Presentation:** 3
**Contribution:** 3
**Rating:** 8
**Confidence:** 3

**Summary:**

Traditionally in Inverse Reinforcement Learning (IRL), when expert demonstrations are available, a generative and adversarial setup is often used. In this setup, the discriminator’s goal is to learn the distribution of the desired behavior from expert demonstrations, while the generator’s goal is to produce behaviors that fit the distribution implicitly defined by the discriminator. Methods such as GAIL have proven powerful but come with a drawback: the optimization process. In these settings with a generator and discriminator, the optimal solution is a saddle point; however, it is often difficult to achieve due to issues like mode collapse and instability in training dynamics. In this work, the authors propose using diffusion models as an alternative. They achieve this by learning the score of the state distribution from expert demonstrations and then applying the DS divergence, which measures the difference in scores throughout the diffusion process. This metric is then used as the reward function to train the policy. The authors demonstrate impressive results on challenging humanoid tasks, particularly the Humanoid Walk task.

**Strengths:**

Impressive experimental results, clearly demonstrating the benefit from not having a GAN approach for IRL.
Clear motivation and formulation and writing is of good quality.

**Weaknesses:**

I think there are prior work with a similar motivation.  Can you elaborate on how your is novel compared to the following:

https://openreview.net/pdf?id=NN60HKTur2
https://arxiv.org/pdf/2004.09395
https://arxiv.org/abs/2407.00626

**Questions:**

The paragraph at line 081 could be clarified for readability and comprehension.

Could the authors elaborate on the sentence: "We first fit a score function to the expert demonstrations. Then, at each iteration of our algorithm, we fit a score function to the state distribution of the mixture of the preceding policies, before using the combination of these score functions to define a reward function for the policy search step." Specifically, what does "mixture of the preceding policies" mean?

Your explanation of the algorithm here is helpful, but this part could be more precise. For example, does "mixture of preceding policies" refer to averaging the state distributions from previous iterations, or does it imply some other form of combining past policies? It would also improve clarity if you could provide a concrete example or a diagram illustrating how this mixture is constructed and applied in the algorithm. These additions would address potential confusion and make the process more accessible to readers.

I think there can be additional clarifications in the method: Algorithm 1, would be good to define $c^{(k)}$ explicitly either in the algorithm pseudocode or in the accompanying text. Additionally, adding how $c^{(k)}$ relates to the score functions mentioned earlier and if it's connected to the cost function in Eq. 3 would provide a more comprehensive understanding of the algorithm.

---

> ### Author Response · Authors · 2024-11-23
>
> Thank you for your valuable feedback! We appreciate your input and have provided our response to your questions below.
>
> - Can you elaborate on how your is novel compared to the following...
>
> We would like to elaborate on it below:
>
> [1]: This work uses diffusion models to represent policies and aims to learn a relative reward function which, when employed as classifier guidance, converts the base policy into the expert policy. It is relevant to our work as it also applies diffusion models in the context of imitation learning. However, unlike our method, which uses diffusion models with a score-matching loss to model state(-action) distributions and generate reward signals, their focus lies in diffusion policies and learning through classifier guidance. Additionally, their method is offline and is thus likely to suffer compounding error, while imitation learning w/ online interaction is known to address this (see [4], as well as we did in our paper).
>
> [2] also explore score matching for imitation learning but consider a different class of models--energy-based models, so it is different from our focus on diffusion models.
>
> [3] consider the problem from the other way around-—it uses inverse RL, specifically maximum entropy IRL, to train diffusion models. In contrast, our approach leverages diffusion models and its standard regression-based training method to perform IRL. In other words, they use diffusion for policy, while we use diffusion for reward (for imitation).
>
> We have make sure to include [1,2] into the related work section where we have highlighted in red.
>
> - Could the authors elaborate on the sentence...Specifically, what does "mixture of the preceding policies" mean?
>
> "mixture of the preceding policies" means the mixture of all previous generated policies prior to the current round: let's say we are at round $k$, then the mixture is on $\pi_1,\dots,\pi_{k-1}$. The "mixture policy" is executed by first choosing a policy $\pi^i$ uniformly at random from all these policies and then executing the chosen policy. It is trajectory-level mixture instead of action-level. (Precise definition provided in line 378-380)
>
> That sentence means: We have two diffusion models. One of them learns the state distribution of the expert policy, and the other learns the state distribution of the mixture policy. We then use the score functions of these two diffusion models to define the reward function, which is used to train a new policy at the next round.
>
> If this explanation sounds good for you, we will incorporate it into the corresponding section to improve clarity.
>
> - definition of $c^{(k)}$
>
> We have made the definition of $c^{(k)}$ more clear in Eq.(3) in line 295.
>
> ---
>
> [1] Nuti, Felipe, Tim Franzmeyer, and João F. Henriques. "Extracting reward functions from diffusion models." Advances in Neural Information Processing Systems 36 (2024).
>
> [2] Liu, Minghuan, et al. "Energy-Based Imitation Learning." Proceedings of the 20th International Conference on Autonomous Agents and MultiAgent Systems. 2021.
>
> [3] Yoon, Sangwoong, et al. "Maximum Entropy Inverse Reinforcement Learning of Diffusion Models with Energy-Based Models." The Thirty-eighth Annual Conference on Neural Information Processing Systems.
>
> [4] Swamy, Gokul, et al. "Of moments and matching: A game-theoretic framework for closing the imitation gap." International Conference on Machine Learning. PMLR, 2021.

---

> > ### Comment · Reviewer_yQfZ · 2024-11-24
> >
> > Thanks for the update, the explanation makes sense. My questions have been satisfied. Good luck!

---

### Official Review · Reviewer_fJoG · 2024-11-03

**Soundness:** 3
**Presentation:** 2
**Contribution:** 3
**Rating:** 6
**Confidence:** 2

**Summary:**

This paper proposes a new imitation learning method using score-based diffusion models. Specifically, the authors perform score-matching to minimize the difference between the expert’s and learners’  state distributions. In this way, they can perform IL using states only and also state-action pairs. They provide theoretical bounds on sample complexity when score functions are used, and show that they provide tighter bounds than when IPMs or discriminators are used. The empirical results show that their method SMILING performs favorably compared to other baselines.

**Strengths:**

- The paper's contributions seem fair. They apply previously established score matching using diffusion models to IL. There have been many methods that used diffusion models in imitation learning (Section 2) and thus I rate the novelty as low. However, the authors provide new theoretical bounds and comparisons to discriminator-based methods.

- The paper is fairly well-written and straightforward. I did not check for the correctness of the proofs and thus cannot comment.

**Weaknesses:**

- The framing of the method in terms of inverse reinforcement learning confused me. As far as I know, inverse reinforcement learning infers a reward function from the state-action-next state tuples and uses that to learn a policy. However, this method seems to be pure imitation learning where actions are learned directly from states or state-action pairs. Section 2 seems to imply that SMILING does IRL but I  see this method as an imitation learning method (without a discriminator). Can the authors clarify if a reward function is inferred or clarify on how SMILING is indeed an IRL method?
- Some parts of the paper are disconnected with the rest of the paper and might possibly be left over from previous drafts  (e.g. connection to IRL, SAC/DreamerV3, etc.).
   - Lines 267-269: Were SAC and DreamerV3 used in the experiments? The experiments compared against DAC and BC.
- It's unclear how Section 6.3 helps to show that score functions are more expressive than discriminators. The results just seem to exploit a known weakness with adversarial training (mode collapse). Does the expressivity here mean that score matching does not suffer from mode collapse?

**Questions:**

See weaknesses

---

> ### Author Response · Authors · 2024-11-23
>
> Thank you for your valuable feedback! We understand your concerns regarding our paper and would like to provide detailed responses to each of them below.
>
> - They apply previously established score matching using diffusion models to IL. There have been many methods that used diffusion models in imitation learning (Section 2) and thus I rate the novelty as low.
>
>
> We would like to clarify that our work is fundamentally distinct from prior works on diffusion models in either the problem setting or the methodology. We take four representative prior works below as example. More details can be found in our related work section.
>
> 1. Wang et al., “DiffAIL: Diffusion Adversarial Imitation Learning”, AAAI 2024: They insert the score matching loss directly into the f-divergence (specifically JS-divergence) objective of the discriminator. In other words, they are still training some specific discriminator based on JS-divergence. Hence, their method still belongs to the GAIL framework that we are comparing against. **In contrast, we do not train any discriminator at all; instead, we only use score matching (i.e., just regression)**.
>
> We have incoprate their algorithm as a baseline in our experiments; please refer to Figures 2 and 4 in our draft where we show that our algorithm consistently outperform theirs.
>
> 2. Pearce et al., “Imitating Human Behaviour with Diffusion Models”, ICLR 2024: They focus on policy parameterization and directly use a diffusion model to model the policy, which is orthogonal to what we are doing. We do not focus on policy parameterization; instead, we are focused on using a diffusion model to model state(-action) distribution (rather than modeling policies in BC) and generate reward signal for imitation. Note that our policy can have any structure, such as an MLP, and we can even integrate their diffusion-based policy into our framework.
>
> 3. Chen et al., “Diffusion Model-Augmented Behavioral Cloning”, ICML 2024: They propose a Behavior Cloning algorithm with an auxiliary loss function computed from diffusion model. In contrast, we consider inverse RL where diffusion models are used to model state(-action) distributions. So ours differs from theirs in methodology (it is BC vs IRL). Also note that, since their algorithm is BC, it requires expert action while we do not; in addition, BC is likely to suffer compounding error while it is known that IL w/ additional online interaction addresses compounding error (see [1], also as we did in this paper).
>
> [1] Swamy, Gokul, et al. "Of moments and matching: A game-theoretic framework for closing the imitation gap." International Conference on Machine Learning. PMLR, 2021.
>
> 4. Lai et al., "Diffusion-Reward Adversarial Imitation Learning", NeurIPS 2024: They use diffusion model as discriminator and their Diffusion Reward is from the standard JS-divergence objective. So it also belongs to the GAIL framework that we are comparing against just like Wang et al.
>
> In summary, prior works either consider a totally different problem setting (e.g. policy parameterization using diffusion model) or using different methodology--for example, some use the GAN-style discriminator that we are comparing against in the paper; in contrast, we do not train a discriminator at all; instead, we only use score matching (i.e., just regression). This is the central contribution of our paper. Given that no prior work has introduced such an approach, we believe this is a novelty. We will also incorporate a more comprehensive review of related work in the next version to clarify it.
>
> - Can the authors clarify if a reward function is inferred or clarify on how SMILING is indeed an IRL method?
>
> Yes, we do infer a reward (cost) function during training. See Eq. (3): $c^k(s)$ is the cost function. Our algorithm is really minimizing this cost function (see line 4 of our algorithm).
>
> - connection to SAC/DreamerV3 / Were SAC and DreamerV3 used in the experiments?
>
> Our algorithm requires an RL subroutine solver (see line 4 of our algorithm) to minimize the cost function we infer, and SAC and DreamerV3 are exactly the RL solvers we are using. Specifically, in line 4 of our algorithm, SAC or DreamerV3 can be employed to perform this minimization. Notably, our algorithm does not rely on any specific RL subroutine and can work with other suitable alternatives as well.

---

> ### Author Response · Authors · 2024-11-23
>
> - It's unclear how Section 6.3 helps to show that score functions are more expressive than discriminators.
>
> Section 6.3 (now moved to Appendix D.3 to save some space since we added more experiments) is designed to support the key arguments we made in Section 5.1 (now moved to Appendix A): **using score functions, we can employ a less expressive function class to represent a richer class of distributions** than f-divergence- and IPM-based approach, making them more effective in imitation learning.
>
> There we provided an example to illustrate this point. When the target distribution belongs to the exponential family (a common class of distributions, expressed as $p(s) = \exp(w^T \phi(s))/Z$), the score function can represent it easily, as its score function is linear. In contrast, methods based on f-divergence or IPM will require nonlinear discriminators. These methods require more expressive function classes (empirically, larger neural networks), which in turn increase the sample complexity needed for learning.
>
> If you think this part is important in building our claim, we can definitely move this section to the main content.

---

> ### Comment · Reviewer_fJoG · 2024-11-27
>
> I thank the authors for the clarifications and all of the answers made sense. I have raised my score.

---

### Official Review · Reviewer_jpw4 · 2024-11-04

**Soundness:** 3
**Presentation:** 3
**Contribution:** 3
**Rating:** 6
**Confidence:** 4

**Summary:**

This paper addresses the problem of online imitation learning. To this end, the paper proposes an algorithm that matches the learner's distributions to the expert's using a cost function computed by a diffusion model. Theoretical derivations demonstrate the advantages of minimizing the proposed Diffusion Score Divergence. However, the experiment results are currently insufficient to fully support the effectiveness of the proposed approach.

**Strengths:**

**Motivation and intuition**
- The motivation behind the proposed Diffusion Score Divergence is convincing, as diffusion models have demonstrated a strong capability for distribution modeling.

**Theoretical contribution**
- This paper presents solid theoretical contributions, including well-defined assumptions, lemmas, and derivations. Additionally, it discusses related topics such as sample efficiency, second-order bounds and comparisons between score functions and discriminators.

**Clarity**
- The writing is clear overall, providing both theoretical and intuitive explanations. The notations, formulations, and theorems are well-explained and easy to follow.

**Weaknesses:**

**Experiment setup**

The experiments are not sufficient. While the claims look promising, the proposed method only compares two baselines, behavior cloning (BC) and discriminator actor-critic (DAC), which is not sufficiently convincing.

First, recent adversarial imitation learning baselines, such as DiffAIL [1], should be considered.
Although these methods do not explicitly learn reward functions but instead train discriminators, I am not convinced that this work is orthogonal to these previous works. They are feasible in the experimental setup in this paper, which includes limited expert demonstrations and online learner demonstrations.

Second, prior works [2,3] demonstrate that diffusion models are effective policies for offline imitation learning. Although offline methods can suffer from compounding error issues, it is still necessary to compare to one of the above methods to show that the proposed score-matching approach is beneficial when online learner demonstrations are accessible.

**Experiment details**
- The details of the expert policy are missing. Since all experiments are normalized by the performance of the expert and random policies, the authors should report the specific performance values of these policies as reference points; otherwise, the reported results remain unclear.
- The number of environment steps used to train the expert policy is also missing, which makes it difficult to draw conclusions about sample efficiency from the results in Figures 3, 4, and 5. Specifically, online imitation learning aims to leverage expert demonstrations to improve sample efficiency. If the number of training steps for the expert policy is on a similar scale (e.g., 10^6), directly applying reinforcement learning to learn the expert policy could be a simpler solution.

[1] Wang, Bingzheng, et al. "DiffAIL: Diffusion Adversarial Imitation Learning." Proceedings of the AAAI Conference on Artificial Intelligence (AAAI). 2024

[2] Chi, Cheng, et al. "Diffusion policy: Visuomotor policy learning via action diffusion." Robotics: Science and Systems (RSS). 2023

[3] Pearce, Tim, et al. "Imitating human behaviour with diffusion models."  International Conference on Learning Representations (ICLR). 2023

**Questions:**

As noted in the weaknesses section, I have the following questions:

- Are there any results that compare the proposed method to recent adversarial imitation learning baselines, such as DiffAIL?
- Are there any results that compare the proposed method to offline imitation methods that utilize diffusion models?
- Could the authors provide the performance and learning details of both the expert policy and the random policy?

---

> ### Author Response · Authors · 2024-11-23
>
> Thank you for your valuable feedback! We appreciate your comments about our experiments and have conducted additional experiments based on your suggestions. In particular, please find our responses to each of your concerns below.
>
> - The experiments are not sufficient.
>
> We have added empirical comparison to a related algorithm **DiffAIL** [1] and a state-of-the-art algorithm **IQ-Learn** [4]. We ran them on all our tasks. The results are already added in Figures 2 and 4, which suggest our method consistently outperforms all of them.
>
> Regarding [2,3], we would like to clarify that they are truly orthogonal to ours. They focus on policy parameterization by representing the policy as a diffusion model. In contrast, our work centers on imitation learning, focusing on how to match state(-action) occupancies and generate reward signals.
>
> Due to this orthogonality, we could even replace our current MLP-based policy by their diffusion policy to take their advantage; but this is orthogonal to what we are focusing on in this paper.
>
> - the authors should report the specific performance values of these policies as reference points
>
> Thanks for this suggestion! We have included the "unnormalized" versions of experiment results in Appendix D.4 for reference!
>
> - If the number of training steps for the expert policy is on a similar scale (e.g., 10^6), directly applying reinforcement learning to learn the expert policy could be a simpler solution.
>
>
>
> We want to clarify that we focus on imitation learning, where we only have access to expert states and have **no reward information** in both the dataset and the online interaction, so unclear how to do RL here. Actually, we merely use RL to train expert and thus generate demonstrations. In practice, we could imagine them coming from real people as is common in robot learning. In that case, the demonstrations naturally come without reward signals so people usually use imitation learning.
>
> ---
>
> [1] Wang, Bingzheng, et al. "DiffAIL: Diffusion Adversarial Imitation Learning." Proceedings of the AAAI Conference on Artificial Intelligence (AAAI). 2024
>
> [2] Chi, Cheng, et al. "Diffusion policy: Visuomotor policy learning via action diffusion." Robotics: Science and Systems (RSS). 2023
>
> [3] Pearce, Tim, et al. "Imitating human behaviour with diffusion models." International Conference on Learning Representations (ICLR). 2023
>
> [4] Garg, Divyansh, et al. "Iq-learn: Inverse soft-q learning for imitation." Advances in Neural Information Processing Systems 34 (2021): 4028-4039.

---

> > ### Comment · Reviewer_jpw4 · 2024-11-26
> >
> > I thank the authors for their update. Please find the responses to each question below.
> > > We have added empirical comparison to a related algorithm DiffAIL [1] and a state-of-the-art algorithm IQ-Learn [4].
> >
> > Thanks for the update. The additional experiments resolve my concern.
> >
> > > Regarding [2,3], we would like to clarify that they are truly orthogonal to ours. They focus on policy parameterization by representing the policy as a diffusion model. In contrast, our work centers on imitation learning, focusing on how to match state(-action) occupancies and generate reward signals.
> >
> > I agree that utilizing diffusion models as a policy [2, 3] and using them to generate reward signals (SMILING) represent two distinct approaches.
> > However, it would be interesting and important to explore whether SMILING remains beneficial if the MLP policy is replaced by diffusion policies.
> > This could be a potential future work.
> >
> > > We have included the "unnormalized" versions of experiment results in Appendix D.4 for reference!
> >
> > Thanks for the update. The results are helpful.
> >
> > > Actually, we merely use RL to train expert and thus generate demonstrations. In practice, we could imagine them coming from real people as is common in robot learning. In that case, the demonstrations naturally come without reward signals so people usually use imitation learning.
> >
> > This contradicts the authors' statement in lines 467-469: "We use the aforementioned RL solvers to train expert policies and collect demonstrations by running the expert policy for five episodes. Each task has a fixed 1K-step horizon without early termination, and thus the resulting dataset has 5K steps per task."
> > I am well aware that, in real-world scenarios, demonstrations are typically provided by humans. However, if the expert demonstrations in this paper were collected using expert policies trained through RL (e.g., SAC), the authors should include the relevant details to clarify this methodology.
> >
> > **Summary**
> > My only concern is that the expert policy and demonstrations used in this paper are still unclear.

---

> ### Author Response · Authors · 2024-11-26
>
> We are delighted that our experiments have addressed your previous concerns.
>
> In response to your valuable suggestions, we have included additional details on expert training and data collection in our manuscript (see Lines 1353-1363 on Page 26). We will release the expert checkpoints, expert demos, and code upon acceptance.
>
> To summarize, we use RL solvers (SAC for DMC tasks and DreamerV3 for HumanoidBench tasks) to train the expert policies on benchmarks’ ground truth rewards. Once the expert training is complete, we generate expert demonstrations by running the learned expert policy for five episodes for each task. Since each episode has a fixed horizon of 1K steps, the resulting dataset contains $5 \times 1,000 = 5,000$ samples per task. This approach is consistent with prior works on imitation learning (including GAIL and DAC) where the expert demos are also generated by first training a policy using RL w/ the ground truth reward, and then collecting trajectories from the RL policy.
>
> We hope this clarifies our setup and are happy to engage in further discussion.

---

> > ### Comment · Reviewer_jpw4 · 2024-11-27
> >
> > Besides the average total return in line 1357, could you include the number of environment steps for expert training as well?

---

> > > ### Author Response · Authors · 2024-11-27
> > >
> > > Thank you very much for your further feedback! We have added the number of training steps (equivalent to the environment steps in our implementation) on page 26. We have copied it below for your convenience.
> > >
> > > > **Expert Training and Demonstration Collection.**
> > > We use the aforementioned RL algorithms (SAC for DMC tasks and DreamerV3 for HumanoidBench tasks) to train the expert policies. We did not change the hyperparameters for these algorithms, making them the same as those provided in their original implementations in Yarats & Kostrikov (2020); Sferrazza et al. (2024). In other words, these algorithms were directly applied to their respective benchmarks without modification.
> > > >
> > > > The RL algorithms were run until the performance plateaued, with the required number of training steps varying across tasks: ball-in-cup-catch (0.1M steps), cheetah-run (4.8M steps), humanoid-walk (3.4M steps), humanoid-sit (2.7M steps), humanoid-crawl (5.4M steps), and humanoid-pole (16M steps). The resulting expert policies  achieve the following average total return: 972.85 for ball-in-cup-catch, 915.16 for cheetah-run, 875.61 for humanoid-walk, 794.12 for humanoid-sit, 914.31 for humanoid-crawl, and 879.48 for humanoid-pole.
> > > >
> > > > Once the expert training is complete, we generate expert demonstrations by running the learned expert policy for five episodes for each task. Since each episode has a fixed horizon of 1K steps, the resulting dataset contains $5 \times 1,000 = 5,000$ samples per task.
> > >
> > > Please let us know if you have any additional comments or suggestions. We are happy to engage in further discussion.

---

> > > > ### Comment · Reviewer_jpw4 · 2024-11-27
> > > >
> > > > Thank you for the update. My concerns have been addressed, and I have raised my score to 6.

---

### Official Review · Reviewer_TxHS · 2024-11-06

**Soundness:** 3
**Presentation:** 3
**Contribution:** 2
**Rating:** 6
**Confidence:** 4

**Summary:**

This work focuses on imitation learning from observations through the introduction of SMILING (Score-Matching Imitation LearnING). SMILING proposes to match the state distributions of experts and imitators by leveraging a diffusion model.
This study presents theoretical results related to SMILING, specifically regarding first and second-order approximation bounds of suboptimality. Additionally, the authors theoretically demonstrate that score-matching methods can achieve superior data efficiency compared to traditional f-divergence or Integral Probability Metric (IPM) matching methods, particularly under deterministic transition scenarios.

**Strengths:**

This work presents a straightforward yet powerful objective: to match the score functions of expert and imitator state distributions. Despite its simplicity, this objective is strongly supported by theoretical results grounded in plausible assumptions. It is expected that the proposed method, SMILING, will show enhanced data efficiency relative to traditional imitation learning (IL) methods where the expert's (cumulative) cost is nearly deterministic as supported by the analysis in line 368.

**Weaknesses:**

I have main concerns in this study as follows:

**1. Insufficient comparative analysis with other diffusion-based IL methods:**

While this work introduces new theoretical contributions to the score-matching IL objective, the method itself lacks novelty.
Previous studies [A, B, C, D] have already explored integrating diffusion models with imitation learning.
However, the manuscript does not provide a detailed comparison between SMILING and these existing approaches, both conceptually and empirically.
For instance, what are the key distinctions or what is the novelty compared to these methods? Under what conditions does the proposed method offer advantages over similar methods? Such discussions should be provided to clarify the novelty of this work.

This work appears to attempt to distinguish itself from other approaches by addressing the more specific problem of LfO, rather than standard IL. However, the manuscript does not show specific considerations for LfO; the stated objective could readily apply to state-action matching, as demonstrated in experiments (Figure 4).

**2. Weak Experimental Evaluations:**

The experimental evaluation presented also raises concerns. In the manuscript, SMILING is compared only against two obsolete baselines: BC and DAC. It is recommended to include comparisons with recent state-of-the-art IL methods such as ValueDICE [E], IQlearn [F], etc.
Moreover, the absence of comparisons with similar diffusion-based IL methods (e.g. DiffAIL [A], Diffusion-KDE [B], DBC [C], DRAIL [D] ...) further weakens the empirical validation.
Such comparisons are insufficient to establish SMILING as a competitive method.

In addition, the lack of experimental validation on theoretical results makes the paper less compelling.
Given the theoretical emphasis on data efficiency in Section 5, experiments quantifying the relationship between the number of expert trajectories and the converged normalized score are essential for substantiating these claims.
Additionally, providing experimental outcomes in both stochastic and deterministic cost scenarios would enhance the reliability of the theoretical results as well.

**References**

[A] Wang et al., “DiffAIL: Diffusion Adversarial Imitation Learning”, AAAI 2024.

[B] Pearce et al., “Imitating Human Behaviour with Diffusion Models”, ICLR 2024.

[C] Chen et al., “Diffusion Model-Augmented Behavioral Cloning”, ICML 2024.

[D] Lat et al., "Diffusion-Reward Adversarial Imitation Learning", NeurIPS 2024.

[E] Kostrikov et al., "Imitation Learning via Off-Policy Distribution Matching", ICLR 2020.

[F] Garg et al., "IQlearn: Inverse soft-Q Learning for Imitation", NeurIPS 2021.

**Questions:**

Q1. Is the theoretical result specific to state-matching only? or can it also be applied to (state-action)-pair score matching?

Q2. LobsDICE [G] highlights a limitation of the state-matching objective, specifically that $D_f (d_E(s) | d_\pi (s))$ fails when it ignores the directionality of expert trajectories, and they suggest using $D_f (d_E(s,s’) | d_\pi (s,s’))$ instead (see the appendix of [G]).
Can the score-matching objective $D_{DS}$ circumvent these issues?

**Reference**

[G] Kim et al., “LobsDICE: Offline Learning from Observation via Stationary Distribution Correction Estimation”, NeurIPS 2022.

**Details Of Ethics Concerns:**

I have no ethical concerns on this work.

---

> ### Author Response · Authors · 2024-11-23
>
> Thank you for your valuable feedback! We appreciate your comments regarding our experiments and have conducted additional experiments as per your suggestions. Moreover, We have provided a more thorough comparison with related works. Please find our response to each of your concerns below.
>
> - Previous studies [A, B, C, D] have already explored integrating diffusion models with imitation learning.
>
> We would like to clarify that our work is fundamentally distinct from [A] and [D] in the problem setting and [B] and [C] in methodology. We elaborate on this below:
>
> **[A] & [D]**: [A] insert the score matching loss directly into the f-divergence (specifically JS-divergence) objective of the discriminator. Similarly, [D] use diffusion model as discriminator and their Diffusion Reward is from the standard JS-divergence objective. Hence, both of them are still training some specific discriminator based on JS-divergence, and thus, their methods still belong to the GAIL framework that we are comparing against.
>
> In other words, **both [A] and [D] are training a discriminator using JS-divergence objective (so it is still GAIL); in contrast, we do not train a discriminator at all;** instead, we only use score matching (i.e., just regression). This is the central contribution of our paper. Given that no prior work has introduced such an approach, we believe this is a novelty.
>
> In addition, we have incorporated [A] as a baseline in our experiments; please refer to Figures 2 and 4 in our draft where we show that our algorithm consistently outperform theirs.
>
>
> **[B]**: They focus on policy parameterization and directly use a diffusion model to model the policy, which is orthogonal to what we are doing. We do not focus on policy parameterization; instead, we are focused on using a diffusion model to model state(-action) distribution (rather than modeling policies in BC) and generate reward signal for imitation. Note that our policy can have any structure, such as an MLP, and we can even integrate their diffusion-based policy into our framework.
>
> **[C]**: They propose a Behavior Cloning algorithm with an auxiliary loss function computed from diffusion model. In contrast, we consider inverse RL where diffusion models are used to model state(-action) distributions. So ours differs from theirs in methodology (it is BC vs IRL). Also note that, since their algorithm is BC, it requires expert action while we do not; in addition, BC is likely to suffer compounding error while it is known that IL w/ additional online interaction addresses compounding error (see [H], also as we did in this paper).
>
> We have made sure to include these works in the related work section where we have highlighted them in red. We will also incorporate a more comprehensive review of related work in the next version.
>
>
> - SMILING is compared only against two obsolete baselines: BC and DAC. It is recommended to include comparisons with...
>
> Thanks for these suggestions! We have added empirical comparison to related algorithm **DiffAIL** and a recent state-of-the-art algorithm **IQ-Learn**. Also, prior works have shown that **ValueDICE** is at best comparable to IQ-Learn (e.g.,[E], [F]), so we think comparision to IQ-Learn is enough. For these algorithms, we use their official implementation while making sure the neural networks are of same size with ours.
>
> We ran them on all our tasks. The results are added in Figures 2 and 4, which suggest our method consistently outperforms all of them.
>
> - experiments quantifying the relationship between the number of expert trajectories and the converged normalized score are essential for substantiating these claims
>
> Thanks for this suggestions and we found this very reasonable! Hence, we conducted an additional experiment to study the impact of varying the number of expert demonstrations on the performance of the learned policy. The results are presented in Appendix D.2 (specifically in Figures 5 and 6). We observe that SMILING can achieve near-expert-level performance with just 5K expert states (Figure 5), whereas DAC requires 25K to reach similar performance. This indicates that SMILING is more efficient in utilizing expert demonstrations than GAN-style discrimator methods, which support our theoretical results (Theorem 1). If you find this section helpful in supporting our theory, we should move it back in the main content to emphasize it.

---

> ### Author Response · Authors · 2024-11-23
>
> - experimental outcomes in both stochastic and deterministic cost scenarios
>
> We hope the additional experiments we provided in the previous two responses have demonstrated the better efficiency of our algorithm. In addition, we want to emphasize that the experiments we conducted are not entirely deterministic. Cost is not the only source of randomness (also noting that we are not assuming determinisitic cost either, and moreover, our policies are also random). However, as demonstrated by our previous experiments on the relationship between the number of expert demonstrations and return, our algorithm can outperform others with significantly better sample efficiency (Figure 5) despite it is not deterministic.
>
> - Q1. Is the theoretical result specific to state-matching only? or can it also be applied to (state-action)-pair score matching?
>
> The theoretical results can be extended to state-action pairs for sure, as we can imagine an "augmented state" by concatenating the original state and action together. The entire proof remains valid.
>
> - Q2. LobsDICE [G] highlights a limitation of the state-matching objective...
>
> Thank you for bringing this interesting point! In our setting, this issue does not pose any problem.
>
> Our goal is to maximize the return (or specifically, to ensure that the total return of the learned policy matches that of the expert). Recall that we have defined the reward solely based on the state (this seems the best we can do given that we only have expert state, but the following argument will be extendable to learning from states and actions as well). Hence, to maximize return, it suffices to align their state distributions $d(s)$, because for any two policies $\pi_1$ and $\pi_2$, we have:
>
> $V^{\pi^1} - V^{\pi^2} =\sum_s \big(d^{\pi^1}(s) - d^{\pi^2}(s) \big) r(s)$
>
> This implies that if $d(s)$ matches, the total returns will naturally match as well. In the counterexample provided in [G], while the policies may visit states in different order, the state distribution is same. Therefore, matching state distribution is sufficient for matching value, matching trajectory distributions is unnecessary.
>
> If you find this explanation reasonable, we would be happy to include their argument and this clarification in our draft.
>
> ---
>
> [A] Wang et al., “DiffAIL: Diffusion Adversarial Imitation Learning”, AAAI 2024.
>
> [B] Pearce et al., “Imitating Human Behaviour with Diffusion Models”, ICLR 2024.
>
> [C] Chen et al., “Diffusion Model-Augmented Behavioral Cloning”, ICML 2024.
>
> [D] Lat et al., "Diffusion-Reward Adversarial Imitation Learning", NeurIPS 2024.
>
> [E] Chang et al. "Adversarial Imitation Learning via Boosting." ICLR 2024.
>
> [F] Garg et al., "IQlearn: Inverse soft-Q Learning for Imitation", NeurIPS 2021.
>
> [G] Kim et al., “LobsDICE: Offline Learning from Observation via Stationary Distribution Correction Estimation”, NeurIPS 2022.
>
> [H] Swamy et al. "Of moments and matching: A game-theoretic framework for closing the imitation gap." ICML, 2021.

---

> > ### Comment · Reviewer_TxHS · 2024-11-27
> >
> > Thank you for addressing my questions and conducting additional experiments!
> > The author response has resolved most of my concerns; however, I still have a few remaining questions:
> >
> > **1. Advantages of SMILING with DiffAIL-style approaches**
> >
> > Thank you for clarifying the previous diffusion-based IL approaches.
> > Then, what are the key advantages of SMILING compared to DiffAIL-style approaches (i.e., methods that use diffusion models as discriminators)?
> > For example, is the proposed DS divergence-based training objective always superior to discriminator-based objectives? or what conditions does it better than JS divergence? or, does its use offer advantages in terms of computational efficiency?
> >
> > **2. Regarding the Problem Setting: IL from observations**
> >
> > The proposed approach does not appear to be specifically designed for the IL-from-observations setting.
> > Instead, it seems applicable to a more general IL problem, particularly when the underlying cost/reward function is state-dependent.
> > In this context, it seems more natural to frame the work in the broader direction of addressing the general IL problem.
> > Could the authors provide their perspective on this problem setting?

---

> > > ### Author Response · Authors · 2024-11-28
> > >
> > > Thank you for your further feedback! We would like to share our thoughts on the two questions below.
> > >
> > > **1. Advantages of SMILING with DiffAIL-style approaches**
> > >
> > > This is an important question, and we are happy to clarify. We explored the benefits of our method over GAIL-style method (including DiffAIL) in Appendix A. To summarize, we have the following benefits:
> > >
> > > (1) Representational Benefits (Lines 824-845): using score functions, we can employ a less expressive function class to represent a richer class of distributions than these approaches. This means we can use simpler models to capture more complicated distributions when doing IL. This is empirically verified in our experiments in Appendix D.3 where we model both the GAIL-style discriminator and our score function as *linear* functions. The results show that GAIL-style approaches degrade significantly, while our method maintains its performance.
> > >
> > > (2) Sample Complexity Benefits (Lines 847-857): GAIL-style approaches (including DiffAIL) are unlikely to achieve the same sharp sample complexity bounds as established in our Theorem 1. This is also empirically validated in our experiments (see Appendix D.2, particularly Figure 5). This means they would require more expert demos than ours. Also, it is also unclear whether these approaches can avoid compounding errors in the same way our algorithm does.
> > >
> > > **2. Regarding the Problem Setting: IL from observations**
> > >
> > > Yes, our method is applicable to general IL problems no matter learning from observations alone or from both observations and actions. The reason we are specifically considering learning from observation in this paper is twofold:
> > >
> > > (1) Extending a state-only IL algorithm to learn from state-action data is generally straightforward, but the reverse may be nontrivial. For instance, IQ-Learn [2] is designed to learn from state-action data. While it can be extended to learn from state-only data (Section 5.4 in their paper), their state-only algorithm has a different objective and thus cannot inherit the theoretical guarantees. Conversely, adapting a state-only IL algorithm to handle state-action data is more straightforward, since we can imagine an "augmented state" by concatenating the state and action together and then apply the state-only algorithm.
> > >
> > > (2) Learning from observation seems more important in some scenarios. For instance, in robotics, getting observations (e.g., through cameras) is often easier than getting expert actions. In such case, Behavior Cloning, a powerful algorithm, cannot be applied since there is no expert actions. This highlights the need for IL algorithms that learn from observation alone.
> > >
> > > We will make sure to clarify in the next version of the paper that our algorithm works for general IL problems to avoid any confusion.
> > >
> > > ---
> > >
> > > [1] Wang, Bingzheng, et al. "DiffAIL: Diffusion Adversarial Imitation Learning." *Proceedings of the AAAI Conference on Artificial Intelligence*. Vol. 38. No. 14. 2024.
> > >
> > > [2] Garg, Divyansh, et al. "Iq-learn: Inverse soft-q learning for imitation." *Advances in Neural Information Processing Systems* 34 (2021): 4028-4039.

---

> > > > ### Comment · Reviewer_TxHS · 2024-11-28
> > > >
> > > > The author's responses are persuasive, and I no longer have any concerns. I have updated my rating to 6.

---

### Meta-Review · Area_Chair_uxxJ · 2024-12-16

**Metareview:**

All the reviewers unanimously recommend accepting this paper, given its clarity and theoretical and empirical results. Also, the author's rebuttal has sufficiently addressed some concerns and questions raised by the reviewers, such as comparisons to recent works, the importance of learning from observations, and the unclear experimental setup and notations. Consequently, I recommend accepting the paper.

**Additional Comments On Reviewer Discussion:**

The author's rebuttal has sufficiently addressed some concerns and questions raised by the reviewers, such as comparisons to recent works, the importance of learning from observations, and the unclear experimental setup and notations. As a result, four reviewers have raised the score after considering the author's rebuttal.

---

### Decision · Program_Chairs · 2025-01-22

Accept (Poster)